# Computational mechanisms of distributed value representations and mixed learning strategies

Shiva Farashahi[1,2✉] & Alireza Soltani [1✉]

Learning appropriate representations of the reward environment is challenging in the real world where there are many options, each with multiple attributes or features. Despite existence of alternative solutions for this challenge, neural mechanisms underlying emergence and adoption of value representations and learning strategies remain unknown. To address this, we measure learning and choice during a multi-dimensional probabilistic learning task in humans and trained recurrent neural networks (RNNs) to capture our experimental observations. We find that human participants estimate stimulus-outcome associations by learning and combining estimates of reward probabilities associated with the informative feature followed by those of informative conjunctions. Through analyzing representations, connectivity, and lesioning of the RNNs, we demonstrate this mixed learning strategy relies on a distributed neural code and opponency between excitatory and inhibitory neurons through value-dependent disinhibition. Together, our results suggest computational and neural mechanisms underlying emergence of complex learning strategies in naturalistic settings.

[1] Department of Psychological and Brain Sciences, Dartmouth College, Hanover, NH, USA. [2] Center for Computational Neuroscience, Flatiron Institute, Simons Foundation, New York, NY, USA. ✉email: sfarashahi@flatironinstitue.org; alireza.soltani@dartmouth.edu

Successful value-based decision making and learning depend on the brain's ability to encode and represent relevant information and to properly update those representations based on reward feedback from the environment. For example, to be able to learn from an unpleasant reaction to consuming a multi-ingredient meal requires having representations for reward value (i.e., subjective reward experience associated with selection and consumption) and/or predictive value of certain individual ingredients or combinations of ingredients that resulted in the outcome (informative attributes). Learning such informative attributes and associated value representations is challenging because feedback is non-specific and scarce (e.g., stomachache after a meal with combinations of ingredients that may never recur), and thus, it is unclear what attributes or combinations of attributes are important for predicting the outcomes and must be learned[1]. Such learning becomes even more challenging in high-dimensional environments where the set of possible stimuli or choice options grows exponentially as the numbers of attributes, features, and/or their instances increase—the problem referred to as the curse of dimensionality.

Recent studies have shown that human and non-human primates can overcome the curse of dimensionality by learning and incorporating the structure of reward environment to adopt an appropriate learning strategy[2–8]. For example, when the environment follows a generalizable set of rules such that the values of choice options can be inferred from their features or attributes, humans follow a feature-based learning to estimate reward value or predictive value of options based on their features instead of learning about individual options (object-based learning) directly[2–11]. In contrast, lack of generalizable rules shifts learning away from fast but imprecise feature-based learning to slower but more precise object-based learning[3,10].

Despite evidence for adoption of such simple feature-based learning, it is currently unknown whether and how more-complex learning strategies and value representations emerge. Specifically, it is not clear whether in high-dimensional environments, humans and other animals adopt representations involving conjunctions of features to go beyond feature-based learning, stop at feature-based learning, or transition to object-based learning when the environment is stable. From a computational point of view, although feature-based learning can be faster because feature values are updated more frequently than conjunction and object values, higher learning rates for object-based and conjunction-based learning could make these strategies more advantageous.

Multiple reinforcement learning (RL) and Bayesian models have been proposed to explain how animals can learn informative representations at the behavioral level[3,5,6,11–14]. However, all these models assume the existence of certain representations or generative models. For example, there are different proposals for what representations are adopted in connectionist models ranging from extreme localist theories, with single processing unit representations (i.e., grandmother cells)[15–17], to distributed theories with representations that are defined as patterns of activity across a number of processing units[18–20]. Although local representations are easy to interpret and are the closest to naïve reinforcement learning models, the scarcity of reward feedback and large number of options in high-dimensional environments make these models unappealing. In contrast, distributed representations allow for more flexibility, making them plausible candidates for learning appropriate representations in high-dimensional reward environments. Nonetheless, it is currently unknown how multiple value representations and learning strategies emerge over time and what the underlying neural mechanisms are.

We hypothesized that in stable, high-dimensional environments, animals start by learning a simplified representation of the environment before learning more complex representations involving certain combinations or conjunctions of features. This conjunction-based learning would provide an intermediate learning strategy that is faster than object-based learning and more precise than feature-based learning alone. At the neural level, we hypothesized that such mixed feature- and conjunction-based learning relies on a distributed representation across different neural types.

Here, to test these hypotheses and investigate whether and how appropriate value representations and learning strategies are acquired, we examine human learning and choice behavior during a naturalistic task with a multi-dimensional reward environment and partial generalizable rules. Moreover, we train recurrent neural networks (RNNs), which have been successfully used to address a wide range of neuroscientific questions[21–32], to perform our task. We find that participants estimate stimulus-outcome associations by learning and combining estimates of reward probabilities associated with the informative feature followed by those of informative conjunctions, and this behavior is replicated by the trained RNNs. To reveal computational and neural mechanisms underlying the emergence of observed learning strategies, we then apply a combination of representational similarity analysis[33], connectivity pattern analysis, and lesioning of the trained RNNs and moreover, explore alternative network structures. We show that the observed mixed learning strategy relies on a distributed neural code and distinct contributions of excitatory and inhibitory neurons. Additionally, we find that plasticity in recurrent connections is crucial for the emergence of complex learning strategies that ultimately rely on opponency between excitatory and inhibitory populations through value-dependent disinhibition.

## Results

**Learning about informative features and conjunctions of features in multi-dimensional environments.** Building upon our previous work on feature-based vs. object-based learning[3], we designed a multi-dimensional probabilistic learning task (mdPL) that allows study of the emergence of intermediate learning strategies. In this task, human participants learned stimulus-outcome associations by selecting between pairs of visual stimuli defined by their visual features (color, pattern, and shape) followed by a binary reward feedback. Moreover, we asked participants to provide their estimates of reward probabilities for individual stimuli during five bouts of estimation trials throughout the experiment (see Methods section for more details). Critically, the reward probability associated with selection of each stimulus was determined by the combination of its features such that one informative feature and the conjunctions of the other two non-informative features could partially predict reward probabilities (Fig. 1). For example, square-shaped stimuli could be on average more rewarding than diamond-shaped stimuli (i.e., average probability of reward on three shapes were equal to 0.3, 0.5, and 0.7), whereas stimuli with different colors (or patterns) were equally rewarding on average (i.e., average probability of reward for these features was equal to 0.5, 0.5, and 0.5). This example corresponds to shape being the informative feature and color and pattern being the non-informative features. At the same time, stimuli with certain combinations of color and pattern (e.g., solid blue stimuli) could be more rewarding than stimuli with other combinations of color and pattern, making conjunctions of color and pattern to be the informative conjunction (i.e., average probability of reward for different color and

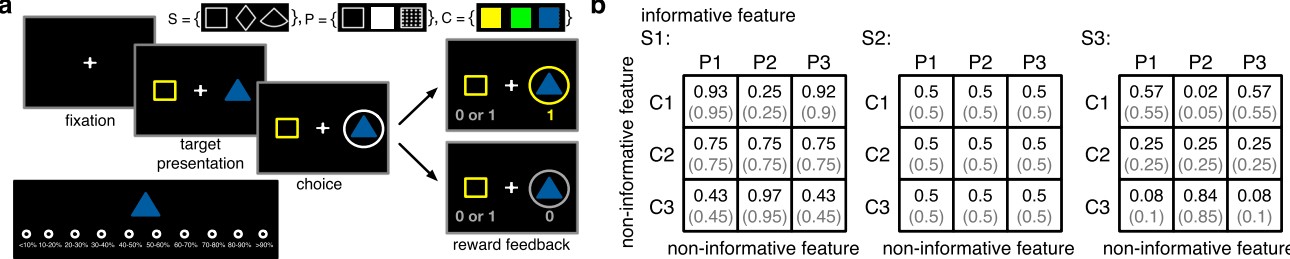

**Fig. 1 Experimental procedure. a** Timeline of a choice trial during the multi-dimensional probabilistic learning task. In each choice trial, the participants chose between two stimuli (colored patterned shapes) and were provided with reward feedback (reward or no reward) for both the chosen and unchosen stimuli. The inset at the top shows the set of all visual features used in the experiment (S: shape; P: pattern; C: color). The inset at the bottom shows the screen during a sample estimation trial. In each estimation trial, the participants provided their estimate about the probability of reward for the presented stimulus by pressing 1 of 10 keys (A, S, D, F, G, H, J, K, L, and;) on the keyboard. **b** Example of reward probabilities assigned to 27 possible stimuli. Stimuli were defined by combinations of three features, each with three instances. Reward probabilities were non-generalizable, such that reward probabilities assigned to all stimuli could not be determined by combining the reward probabilities associated with their features or conjunctions of their features. Numbers in parentheses demonstrate the actual probability values used in the experiment due to limited number of trials. For the example schedule, the shape was on average informative about reward (average probability of reward on three shapes were equal to 0.3, 0.5, and 0.7). Although pattern and color alone were not informative, the conjunction of these two non-informative features was on average informative about the reward. Each participant was randomly assigned to a condition where the informative feature was either pattern or shape.

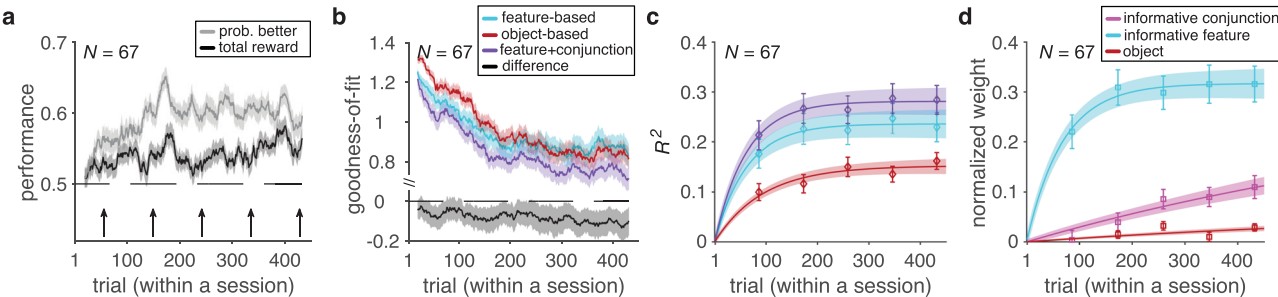

**Fig. 2 Evidence for adoption of mixed feature- and conjunction-based learning. a** Time course of performance and learning during the experiment. Plotted are the average total harvested reward and probability of selecting the better stimulus (i.e., stimulus with higher probability of reward) in a given trial within a session of the experiment. The running average over time is computed using a moving box with the length of 20 trials. The shaded areas indicate ±s.e.m., and the dashed line shows chance performance. **b** Plotted is the goodness-of-fit based on the average AIC per trial, $AIC_p$, for the feature-based model, object-based model, and the best mixed feature- and conjunction-based ($F+C_1$) model. The smaller value corresponds to a better fit. The black curve shows the difference between the goodness-of-fit for the $F+C_1$ and feature-based models. **c** The plot shows the time course of explained variance ($R^2$) in participants' estimates based on different GLMs. Color conventions are the same as in panel **b** with cyan, red, and purple curves representing $R^2$ based on the feature-based, object-based, and $F+C_1$ models, respectively. The solid line is the average of fitted exponential function to each participant's data and the shaded areas indicate ±s.e.m. of the fit. **d** Time course of adopted learning strategies measured by fitting participants' estimates of reward probabilities using a stepwise GLM. Plotted is the normalized weight of the informative feature, informative conjunction, and object (stimulus identity) on reward probability estimates. Error bars indicate s.e.m. The solid line is the average of fitted exponential function to each participant's data, and the shaded areas indicate ±s.e.m. of the fit. Source data are provided as a Source Data file.

pattern conjunction instances were equal to [0.63, 0.27, 0.63; 0.5, 0.5, 0.5; 0.37, 0.74, 0.37]).

Analyzing participants' performance and choice behavior suggested that most participants understood the task and learned about stimuli while reaching their steady-state performance in about 150 trials. This was evident from the average total harvested reward as well as the probability of choosing the better stimulus (i.e., stimulus with higher probability of reward) in each trial over the course of the experiment (Fig. 2a). To identify the learning strategy adopted by each participant, we fit individual participants' choice behavior using 24 different reinforcement learning (RL) models that relied on feature-based, object-based, mixed feature- and conjunction-based, or mixed feature- and object-based learning strategies (see Methods section). By fitting participants' choice data and computing AIC and BIC per trial ($AIC_p$ and $BIC_p$; see Methods section) as different measures of goodness-of-fit[10,34], we found that one of the mixed feature- and conjunction-based models provided the best overall

fit for all data and throughout the experiment (Supplementary Table 1, Fig. 2b, and Supplementary Fig. 1). In this model (the $F+C_1$ model), the decision maker updates the estimated reward probabilities associated with the informative feature and the informative conjunction of the selected stimulus after each feedback while forgetting reward probabilities associated with the unchosen stimulus (by decaying those values toward 0.5). As a result, this mixed feature- and conjunction-based model was able to learn quickly without compromising precision in estimating reward probabilities (Supplementary Fig. 2). Consistent with this, we found that the difference in the goodness-of-fit between this model and the second-best model (feature-based model) increased over time (Fig. 2b), pointing to more use of the mixed learning strategy by the participants.

In addition to choice trials, we also examined estimation trials to determine learning strategies adopted by individual participants over the course of the experiment. To that end, we used three separate generalized linear models (GLMs) to fit

 **3**

participants' estimated reward probabilities associated with each stimulus based on the predicted reward probabilities using different learning strategies (see Methods section). The explained variance from different GLMs confirmed that the $F + C_1$ model captured participants' estimates the best (Fig. 2c).

Although the above analyses of choice behavior and estimation trials confirmed that participants ultimately adopted a mixed (feature- and conjunction-based; $F + C$) strategy, they also showed a small discrepancy. More specifically, toward the end of the session, the fit of choice behavior by the object-based model becomes marginally better than that of the feature-based model (Fig. 2b), but this is not the case for the explained variance in estimated reward probabilities (Fig. 2c). We found that this discrepancy was mainly due to similarity of object-based values to predicted values based on the $F + C_1$ model (spearman correlation; $\rho = 0.86$, $P = 8.3 \times 10^{-9}$), making the object-based model to fit choice behavior better than the feature-based model as participants progress through the experiment. To demonstrate this directly, we ran additional analyses to show that the object-based strategy does not capture more variance than the feature-based strategy. To that end, we used stepwise GLM to fit estimated reward probabilities associated with each stimulus based on the actual reward probabilities (object-based) and the predicted reward probabilities using the informative feature and conjunction. We found that the predicted values based on the informative feature explained most variance followed by the predicted values based on the informative conjunction. Moreover, adding object-based values did not significantly increase the explained variance of estimated reward probabilities beyond a model that included both feature- and conjunction-based values (median ± IQR = 0.9 ± 1.0%; two-sided sign-rank test, $P = 0.12$, $d = 0.48$, $N = 67$). It is worth noting that unlike fitting of choice behavior, which is done using different models separately, stepwise GLM does not suffer from the similarity of object values to predicted values based on the $F + C_1$ model. Together, these results demonstrate that the influence of object-based strategy on estimated reward probabilities did not increase over time and that the observed improved fit of object-based relative to feature-based model was a byproduct of the similarity of object values to predictions of $F + C_1$ model (i.e., the best model).

In addition, the time course of extracted normalized weights for the informative feature regressor and the informative conjunction regressor in a stepwise GLM suggested that participants assigned larger weight to the informative feature and learned the informative feature followed by the informative conjunction. More specifically, we fit the time course of extracted normalized weights to estimate the time constant at which these weights reached their asymptotes (see Eq. 3 in Methods section). We found that the time constant of increase in the weight of informative feature (median ± IQR = 69.3 ± 35.9) was an order of magnitude smaller than the time constant of increase in the weight of informative conjunction (median ± IQR = 985.1 ± 49.0; two-sided sign-rank test, $P = 3.24 \times 10^{-5}$, $d = 0.73$, $N = 67$; Fig. 2d), indicating much faster learning of the informative feature compared with the informative conjunction.

These results are not trivial even though feature-based learning should be learned faster than conjunction-based learning due to more frequent updating of feature than conjunction values. This is because larger learning rates for conjunction-based than feature-based learning can compensate for more frequent updates of feature values. To demonstrate this point, we compared accuracy of different learning strategies during the first 50 trials of the experiment. Specifically, we simulated RL models based on feature-based, conjunction-based, mixed feature- and conjunction-based, or object-based strategy (with decay of values for the unchosen stimulus) and computed error in estimation of reward

probabilities for these models (Supplementary Fig. 3). We found that the superiority of feature-based over conjunction-based strategy depends on the choice of the learning rates and the decay rate. More specifically, early in the learning, a conjunction-based model with a large learning rate exhibits a smaller average squared error in predicting reward probabilities than that of a feature-based learner with a small learning rate, whereas a conjunction-based learner with a small learning rate is less accurate than a feature-based learner with a large learning rate (Supplementary Fig. 3a). Moreover, parameter space for which the feature-based learner is more accurate increases with larger decay rates (Supplementary Fig. 3b, c). Similarly, an object-based learner's accuracy early in the experiment can be better than that of the mixed feature- and conjunction-based learner and the feature-based learner, depending on the learning and decay rates (Supplementary Fig. 3d–i). Together, these simulation results illustrate that the advantage and thus the adoption of certain learning strategies could greatly vary, making our behavioral findings non-trivial.

We also analyzed data from the excluded participants. However, this analysis did not provide any evidence that excluded participants adopted learning strategies qualitatively different from those used by the remaining participants (Supplementary Fig. 4). Instead, it showed that excluded participants simply did not learn the task. Together, these results demonstrate that our participants were able to learn more complex representations of reward value (or predictive value) over time and combined information from these representations with simple representation of individual features to increase their accuracy without slowing down learning.

**Direct evidence for adoption of more complex learning strategies.** To confirm our results on adopted learning strategies more directly, we also used choice sequences to examine participants' response to different types of reward feedback (reward vs. no reward). To that end, we calculated differential response to reward feedback by computing the difference between the tendency to select a feature (or a conjunction of features) of the stimulus that was selected on the previous trial and was rewarded vs. when it was not rewarded (see Methods section, Supplementary Fig. 5). Our prediction was that participants who learned about the informative feature and informative conjunction, as assumed in the $F + C_1$ model, should exhibit positive differential response for the informative feature and informative conjunction (but not non-informative features and conjunctions). In contrast, differential response would be positive for the informative feature (and possibly the non-informative features) for participants who only learned about individual features and not their conjunctions. We used our measure of goodness-of-fit to determine whether a participant adopted feature-based or mixed feature- and conjunction-based learning (62 out of 67 participants). We did not calculate differential response for the small minority of participants (5 out of 67) who adopted the object-based strategy.

As expected, we found that differential response for informative conjunction and informative feature was overall positive for participants who adopted the best mixed feature- and conjunction-based learning strategy (the $F + C_1$ model) (two-sided sign-rank test; informative feature: $P = 8 \times 10^{-4}$, $d = 0.59$, $N = 41$; informative conjunction: $P = 0.029$, $d = 0.19$, $N = 41$). For these participants, differential response for the non-informative features and the non-informative conjunctions were not distinguishable from 0 (two-sided sign-rank test; non-informative features: $P = 0.23$, $d = 0.058$, $N = 41$; non-informative conjunctions: $P = 0.47$, $d = 0.056$, $N = 41$). We also found that differential response for the informative feature was

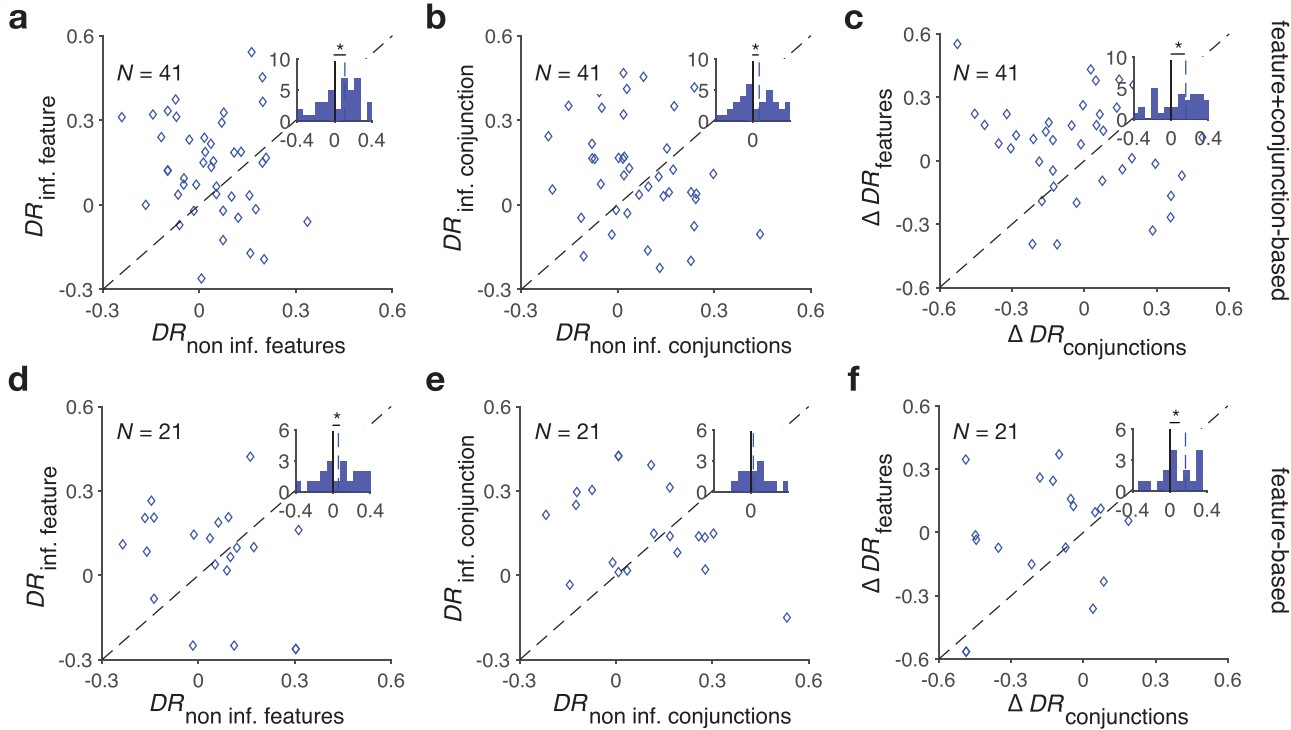

**Fig. 3 Direct evidence for adoption of mixed feature- and conjunction-based learning strategy. a** Plot shows differential response for the informative feature vs. differential response for the non-informative features for participants whose choice behavior was best fit by the $F+C_1$ model. The inset shows the histogram of the difference between differential response of the informative and non-informative features. The dashed line shows the median values across participants, and the asterisk indicates the median is significantly different from 0 (two-sided sign-rank test; $P = 0.013$). **b** Plot shows differential response for the informative conjunction vs. differential response for the non-informative conjunctions for the same participants. The inset shows the histogram of the difference between differential response of the informative and non-informative conjunctions (two-sided sign-rank test; $P = 0.031$). **c** Plot shows the difference between differential response for the informative and non-informative features vs. the difference between differential response for the informative and non-informative conjunctions. The inset shows the histogram of the difference between the aforementioned differences (two-sided sign-rank test; $P = 0.025$). **d–f** Similar to **a–c** but for participants whose choice behavior was best fit by the feature-based model (two-sided sign-rank test; features: $P = 0.005$, conjunctions: $P = 0.15$, difference: $P = 0.045$). Source data are provided as a Source Data file.

larger than that of the non-informative features (two-sided sign-rank test; $P = 0.013$, $d = 0.41$, $N = 41$; Fig. 3a). Similarly, differential response for the informative conjunction was larger than that of the non-informative conjunctions (two-sided sign-rank test; $P = 0.031$, $d = 0.38$, $N = 41$; Fig. 3b). Finally, the difference between differential response for the informative and non-informative features was larger than the difference between differential response for the informative and non-informative conjunctions (two-sided sign-rank test; $P = 0.025$, $d = 0.23$, $N = 41$; Fig. 3c).

In contrast, for participants who adopted the feature-based strategy, only differential response for the informative feature was significantly larger than zero (two-sided sign-rank test; informative feature: $P = 0.022$, $d = 0.37$, $N = 21$; non-informative features: $P = 0.09$, $d = 0.25$, $N = 21$) with differential response for the informative feature being larger than that of non-informative features (two-sided sign-rank test; $P = 0.005$, $d = 0.43$, $N = 21$; Fig. 3d). Moreover, for these participants, differential response for either the informative conjunction or non-informative conjunctions was not distinguishable from zero (two-sided sign-rank test; informative conjunction: $P = 0.27$, $d = 0.17$, $N = 21$; non-informative conjunctions: $P = 0.08$, $d = 0.41$, $N = 21$), and differential response for the informative conjunction was not distinguishable from that of non-informative conjunctions (two-sided sign-rank test; $P = 0.15$, $d = 0.12$, $N = 21$; Fig. 3e). Finally, the difference between differential response for the informative

and non-informative features was larger than the difference between differential response for the informative and non-informative conjunctions (two-sided sign-rank test; $P = 0.045$, $d = 0.31$, $N = 41$; Fig. 3f). Together, these illustrate consistency between differential response analysis and fitting of choice behavior.

Finally, to illustrate the adoption of the mixed learning strategy in a completely model-independent manner, we also compared differential response across all participants. Results of this analysis revealed that participants responded differently to reward vs. no reward depending on the informativeness of both features and conjunctions of the selected stimulus in the previous trial. On the one hand, differential response of the informative feature was significantly larger than zero (two-sided sign-rank test; $P = 10^{-3}$, $d = 0.61$, $N = 67$; Supplementary Fig. 6a), whereas differential response of the non-informative features was not significantly different than zero (two-sided sign-rank test; $P = 0.09$, $d = 0.28$, $N = 67$; Supplementary Fig. 6b). Similarly, differential response of the informative conjunction but not non-informative conjunctions was significantly larger than zero (two-sided sign-rank test; informative conjunction: $P = 2.7 \times 10^{-3}$, $d = 0.65$, $N = 67$; non-informative conjunctions: $P = 0.11$, $d = 0.21$, $N = 67$; Supplementary Fig. 6c, d).

Overall, our experimental results demonstrate that when learning about high-dimensional stimuli, humans adopt a mixed learning strategy that involves learning about the informative

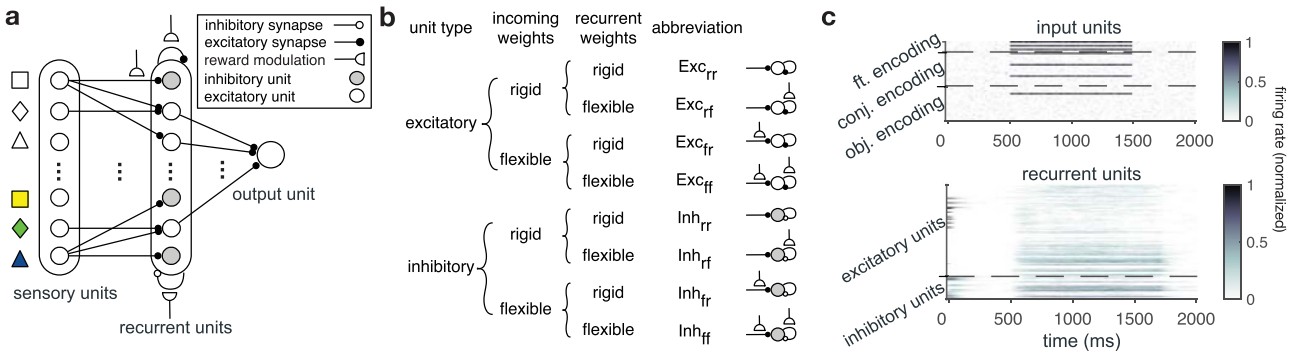

**Fig. 4 Architecture of the RNN models. a** The models consist of three layers with different types of units mimicking different populations of neurons: sensory units, recurrent units, and an output unit. The recurrent units ($N = 120$) included both excitatory and inhibitory populations that receive input from 63 sensory populations encoding individual features ($N = 9$), conjunctions of features ($N = 27$), and object-identity of each stimulus ($N = 27$). Among the recurrent populations only the excitatory recurrent populations project to the output population. Half of the connections from sensory populations and the connections between recurrent populations were endowed with reward-dependent plasticity. **b** Based on the type of populations and uniform presence/absence of reward-dependent plasticity in the connections to and between recurrent populations, these populations could be grouped into eight disjoint populations: $Exc_{rr}$ and $Inh_{rr}$ corresponding to populations with no plastic sensory or recurrent connections (rigid weights indicated by subscript r); $Exc_{fr}$ and $Inh_{fr}$ corresponding to populations with plastic sensory input only (flexible weights indicated by subscript f); $Exc_{rf}$ and $Inh_{rf}$ corresponding to populations with plastic recurrent connections only; and $Exc_{ff}$ and $Inh_{ff}$ corresponding to populations with plastic sensory input and plastic recurrent connections. **c** Activity of the sensory and recurrent populations in an example trial. Upon presentation of a stimulus, three feature-encoding, three conjunction-encoding, and one object-identity encoding populations become active and give rise to activity in the excitatory and inhibitory recurrent populations. Source data are provided as a Source Data file.

feature as well as the informative conjunction of the non-informative features. The informative feature is learned quickly and is slowly followed by learning about the informative conjunction, indicating the gradual emergence of more complex representations over time.

**RNNs adopt intermediate learning strategies similar to human participants.** To account for our experimental observations and gain insight into computational and neural mechanisms underlying the emergence of different learning strategies and value representations, we constructed and trained RNNs to perform our task (Fig. 4). Specifically, we used biologically inspired recurrent networks of point excitatory and inhibitory populations (with 4 to 1 ratio) endowed with plausible reward-dependent Hebbian learning rules[35–37]. Recurrent design was chosen to ensure that the network can demonstrate long-term complex dynamics while learning from reward feedback.

We trained these RNNs in two steps to mimic realistic agents. First, we employed the stochastic gradient descent (SGD) method to train RNNs to learn input-output associations for estimating reward probabilities of the 27 three-dimensional stimuli used in our task, and then we used the trained RNNs to perform our task. For each training session of 270 trials, reward probabilities were randomly assigned to different stimuli to enable the network to learn a universal solution for learning reward probabilities in multi-dimensional environments with different levels of generalizability (i.e., how well reward probabilities based on features and conjunctions predict actual reward probabilities associated with different stimuli). This first training step allowed the networks to learn a general task of learning and estimating reward probabilities without overfitting (Supplementary Fig. 7). Moreover, training in different environments also resulted in natural variability in the trained networks' connectivity, mimicking participant's individual variability in our experiment.

In the second step (simulation of the actual experiment), we stopped SGD and simulated the behavior of the trained RNNs in a session of learning task with reward probabilities used in our experiment. In this step, only connections endowed with the plasticity mechanism were modulated after receiving reward feedback in each trial. The overall task structure used in these simulations was similar to our experimental paradigm with a simple modification where only one stimulus was shown in each trial and the network had to learn the reward probability associated with that stimulus. By such simplification, we assume that there is no effect across chosen and unchosen stimuli, which is supported by the results of fitting participants' choice behavior (Supplementary Table 1). Using this approach, we also avoided complexity related to decision-making processes and mainly focused on learning aspects of the task. In summary, in the first step, RNNs learned to perform a general multi-dimensional reward learning task, whereas in the second step, the trained RNNs were used to perform our task.

We first confirmed that the networks' estimate of reward probabilities during the simulation of our task matched that of the human participants (Fig. 5a). Moreover, we also examined population activity dynamics at different time points in the session to test whether the networks' activity reflected learned reward probabilities (see Methods section). We found that the trajectory of population response projected on the three principal components was not distinguishable at the beginning of the session, whereas this response diverged according to reward probabilities as the network learned these values through reward feedback (Supplementary Fig. 8).

Next, we utilized three separate GLMs to fit reward probability estimates in order to identify learning strategies adopted by the trained RNNs during the course of the experiment. The explained variance of fit of RNNs' estimates confirmed that similar to our human participants, estimates of RNNs over time were best fit by the $F + C_1$ model (Fig. 5b). Moreover, the extracted normalized weights for the informative feature and informative conjunction in a stepwise GLM suggest that similar to our human participants, RNNs learn the informative feature followed by the informative conjunction (Fig. 5c). This was reflected in the time constant of increase in the weight of the informative feature (median ± IQR = 80.12 ± 68.44) being significantly smaller than that of the informative conjunction (median ± IQR = 200.75 ± 107.19;

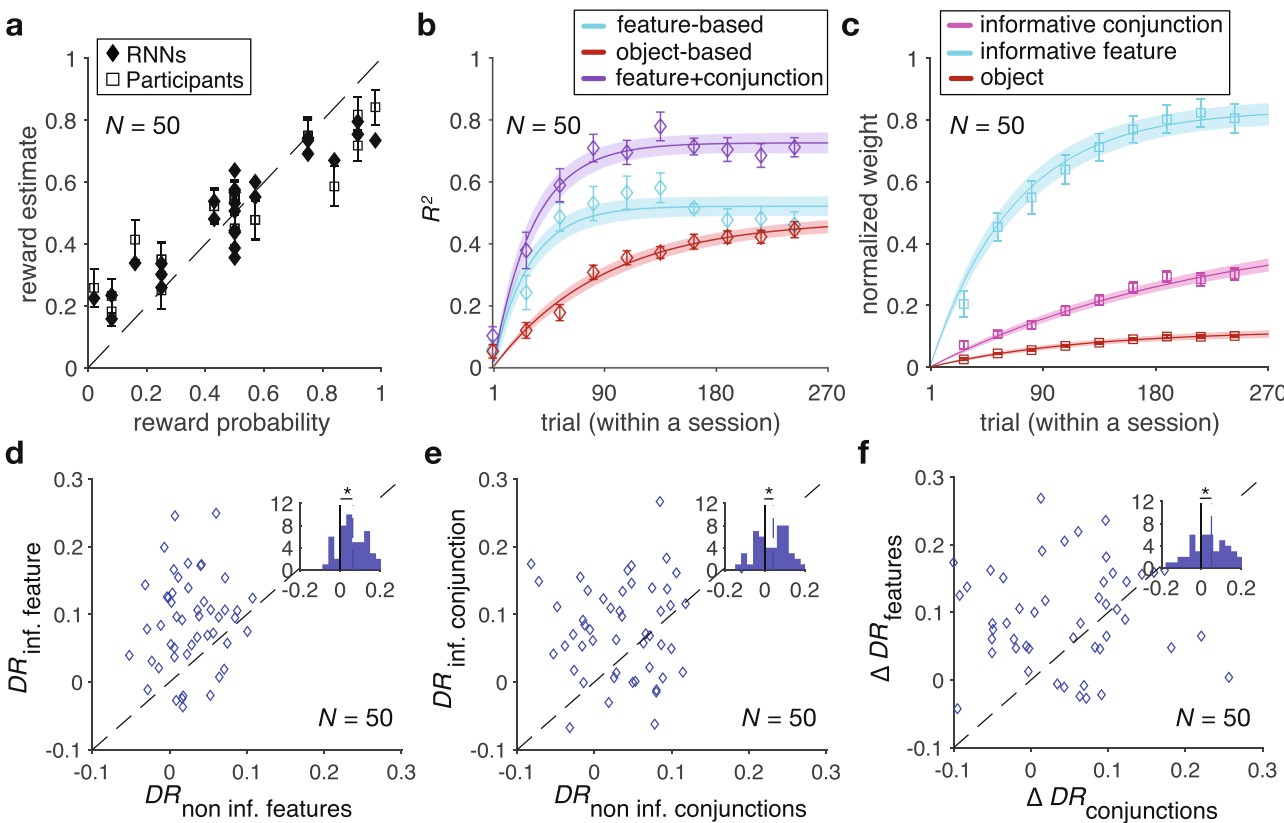

**Fig. 5 RNNs can capture main behavioral results. a** Plotted are average estimates at the end of a simulated session of our learning task for $N = 50$ instances of the simulated RNNs and the average value of participants' reward estimate vs. actual reward probabilities associated with each stimulus (each symbol represents one stimulus). Error bars represent s.e.m., and the dashed line is the identity. **b** The plot shows the time course of explained variance ($R^2$) in RNNs' estimates based on different GLMs. Error bars represent s.e.m. The solid line is the average of exponential fits to RNNs' data, and the shaded areas indicate ±s.e.m. of the fit. **c** Time course of adopted learning strategies measured by fitting the RNNs' output using a stepwise GLM. Plotted is the normalized weight of the informative feature, informative conjunction, and object (stimulus identity) on reward probability estimates. Error bars represent s.e.m. The solid line is the average of exponential fits to RNNs' data, and the shaded areas indicate ±s.e.m. of the fit. **d** Plot shows differential response for the informative feature vs. differential response for the non-informative features in the trained RNNs. The inset shows the histogram of the difference between differential response of the informative and non-informative features. The dashed line shows the median values across all trained RNNs, and an asterisk indicates the median is significantly different from 0 (two-sided sign-rank test; $P = 0.013$). **e** Similar to **d** but for the informative and non-informative conjunctions (two-sided sign-rank test; $P = 1.4 \times 10^{-6}$). **f** Plot shows the difference between differential response for the informative and non-informative features vs. the difference between differential response for the informative and non-informative conjunctions. The inset shows the histogram of the difference between the aforementioned differences (two-sided sign-rank test; $P = 0.018$). Source data are provided as a Source Data file.

two-sided sign-rank test, $P = 0.008$, $d = 0.36$, $N = 50$). Consistent with human participants, RNNs' predicted values based on the informative feature explained most variance followed by the predicted values based on the informative conjunction. Moreover, adding object-based values did not significantly increase the explained variance of estimated reward probabilities beyond a model that included both feature- and conjunction-based values (median ± IQR = 5.7 ± 3.0%, two-sided sign-rank test, $P = 0.08$, $d = 0.96$, $N = 50$).

To avoid overfitting, we did not fit RNNs to choice behavior directly[38]. Nonetheless, to confirm that the trained RNNs can produce choice data compatible with human participants, we fit choice behavior of the trained RNNs using various RL models. To that end, we added a decision layer (using a logistic function) to the output layer of RNNs to generate choice based on the presentation of a pair of stimuli on each trial and found that the mixed feature- and conjunction-based model provided the best overall fit across all trained RNNs (Supplementary Table 2).

Moreover, we also calculated differential response of the trained RNNs using their estimates of reward probabilities in each trial. This analysis revealed results similar to our human

participants. Specifically, we found that in the trained RNNs, differential responses for the informative feature and the informative conjunction were both positive (two-sided sign-rank test; informative feature: $P = 0.03$, $d = 0.28$, $N = 50$; informative conjunction: $P = 0.04$, $d = 0.23$, $N = 50$). In contrast, differential response for the non-informative features and the non-informative conjunctions were not distinguishable from 0 (two-sided sign-rank test; non-informative features: $P = 0.11$, $d = 0.14$, $N = 50$; non-informative conjunctions: $P = 0.13$, $d = 0.09$, $N = 50$). In addition, differential response for the informative feature was larger than that of the non-informative features (two-sided sign-rank test; $P = 0.013$, $d = 0.41$, $N = 50$; Fig. 5d), and differential response for the informative conjunction was larger than that of the non-informative conjunctions (two-sided sign-rank test; $P = 1.4 \times 10^{-6}$, $d = 0.48$, $N = 50$, Fig. 5e). Finally, similar to our experimental results, the difference between differential response for the informative and non-informative features was larger than the difference between differential response for the informative and non-informative conjunctions (two-sided sign-rank test; $P = 0.018$, $d = 0.18$, $N = 50$; Fig. 5f). Together, these results illustrate that the trained RNNs can qualitatively replicate behavior of human participants.

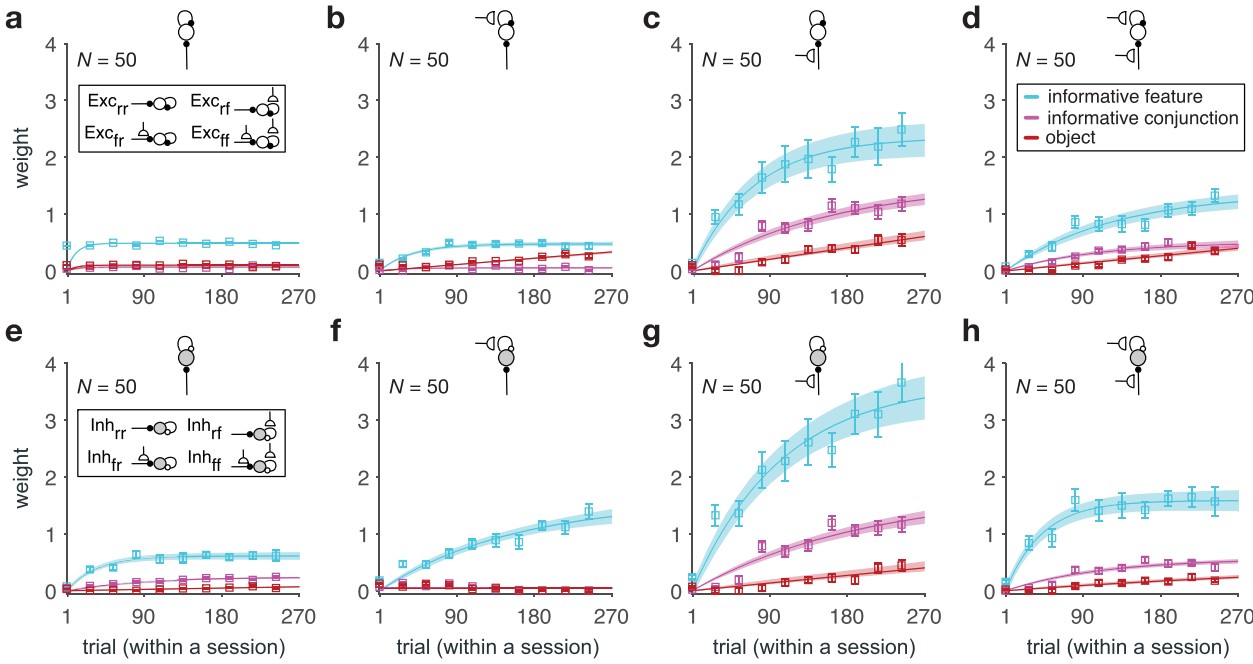

**Fig. 6 Response of different types of recurrent populations show differential degrees of similarity to reward probabilities based on different learning strategies. a–d** Plotted are the estimated weights for predicting the response dissimilarity matrix of different types of recurrent populations (indicated by the inset diagrams explained in Fig. 4b) using the dissimilarity of reward probabilities based on the informative feature, informative conjunction, and object (stimulus identity). Error bars represent s.e.m. The solid line is the average of fitted exponential functions to RNNs' data, and the shaded areas indicate ±s.e.m. of the fit. **e–h** Same as **a–d** but for inhibitory recurrent populations. Dissimilarity of reward probabilities of the informative feature can better predict dissimilarity of response in inhibitory populations, whereas dissimilarity of reward probabilities of the objects can better predict dissimilarity of response in excitatory populations. Source data are provided as a Source Data file.

Together, these results illustrate that our proposed RNNs with biophysically realistic features are able to replicate our experimental findings and exhibit transition between learning strategies over the course of the experiment. Next, we examined the response of the trained RNNs' units to identify the neural substrates underlying different learning strategies and their emergence over time.

**Learning strategies are reflected in the response of different neural types.** To investigate value representations that accompany the evolution of learning strategies observed in our experiment, we applied representational similarity analysis[33] to the response of populations in the trained RNNs. More specifically, we examined how dissimilarity in the response of recurrent populations (response dissimilarity matrix) can be predicted based on the dissimilarity of reward probabilities calculated according to different learning strategies (reward probability dissimilarity matrices; see Methods section for more details). To that end, we used GLMs to estimate the normalized weights of the reward probability dissimilarity matrices in predicting the response dissimilarity matrix (Supplementary Fig. 9). Using this method, we were able to quantify how much representations in recurrent populations reflect or accompany a particular learning strategy.

We found that recurrent populations with plastic sensory input ($Exc_{fr}$, $Exc_{ff}$, $Inh_{fr}$, $Inh_{ff}$) show a strong but contrasting response to reward probabilities associated with stimuli and their features (Fig. 6). Importantly, dissimilarity of reward probabilities based on the informative feature could better predict dissimilarity of response in the inhibitory populations with plastic sensory input ($Inh_{fr}$, $Inh_{ff}$; Fig. 6g, h) compared to the excitatory populations with plastic sensory input ($Exc_{fr}$, $Exc_{ff}$; Fig. 6d). This was reflected

in the difference between the weights of the informative feature for these inhibitory and excitatory populations being significantly larger than 0 (median ± IQR = 0.33 ± 0.19; two-sided sign-rank test, $P = 0.006$, $d = 1.05$, $N = 50$). In contrast, dissimilarity of reward probabilities based on objects could better predict dissimilarity of response in the excitatory populations with plastic sensory input ($Exc_{fr}$ and $Exc_{ff}$) (Fig. 6c, d) compared to the inhibitory populations with plastic sensory input ($Inh_{fr}$ and $Inh_{ff}$) (Fig. 6g, h). This was evident from the difference between the weights of the reward dissimilarity matrix based on objects for these excitatory and inhibitory populations being larger than 0 (median ± IQR = 0.06 ± 0.07; two-sided sign-rank test, $P = 0.002$, $d = 1.32$, $N = 50$).

Finally, we did not find any significant difference between how dissimilarity of reward probabilities based on the informative conjunction predicts dissimilarity in the response of inhibitory and excitatory populations. The difference between the weights of the informative conjunction for inhibitory and excitatory populations was not significantly different from 0 (median ± IQR = −0.004 ± 0.08; two-sided sign-rank test; $P = 0.53$, $d = 0.48$, $N = 50$). However, we found that dissimilarity of reward probabilities based on the informative conjunction can better predict the dissimilarity of response in recurrent populations with plastic sensory input only ($Exc_{fr}$ and $Inh_{fr}$) (Fig. 6c, g) compared to recurrent populations with plastic sensory input and plastic recurrent connections ($Exc_{ff}$ and $Inh_{ff}$) (Fig. 6d, h). This was reflected in the difference between the weights of the informative conjunction for these populations being larger than 0 (median ± IQR = 0.59 ± 0.19; two-sided sign-rank test; $P = 0.001$, $d = 1.94$, $N = 50$).

Together, these results demonstrate distinct contributions of excitatory and inhibitory neurons to different learning strategies and their accompanying value representations and thus, provide a

few predictions about the representations of reward value (or predictive value) by different neural types. First, only neurons with plastic sensory input exhibit value representations compatible with the evolution of learning strategies observed in our task. Second, there is a competition or opponency between representations of object and feature values by excitatory and inhibitory neurons, respectively: excitatory neurons better represent object values, whereas inhibitory neurons better represent feature values. Finally, neurons with only plastic sensory input exhibit more pronounced representations of conjunction values. To better understand the roles of different neural types in learning, we next examined how the different value representations emerge over time.

**Connectivity pattern reveals distinct contributions of excitatory and inhibitory neurons**. To study how different value representations emerge in the trained RNNs, we probed connection weights at the end of the training step and examined how these weights were modulated by reward feedback during the simulation of our task. We refer to the weights at the end of the training step as naïve weights because at that point, the network has not been exposed to the specific relationship between reward probabilities of stimuli and their features and/or conjunctions in our task. These naïve weights are important because they reveal the state of connections due to learning in high-dimensional environments with different levels of generalizability (Supplementary Fig. 10). Moreover, these weights determine the activity of recurrent populations that influences subsequent changes in connections due to reward feedback in our task.

We found that after training in high-dimensional environments with different levels of generalizability, feature-encoding populations were connected more strongly to the inhibitory populations with plastic sensory input (Inh$_{fr}$ and Inh$_{ff}$) (Fig. 7a). Specifically, the average value of naïve weights from the feature-encoding populations to the inhibitory populations with plastic sensory input was significantly larger than the average value of naïve weights from feature-encoding populations to other populations (median ± IQR = 0.07 ± 0.02, two-sided sign-rank test; adjusted $P = 1.5 \times 10^{-3}$, $d = 1.21$, $N = 50$). In contrast, object-identity encoding populations were connected more strongly to the excitatory populations with plastic sensory input (Exc$_{fr}$ and Exc$_{ff}$). Specifically, the average value of naïve weights from the object-encoding populations to the excitatory populations with plastic sensory input was significantly larger than the average value of naïve weights from object-encoding populations to other populations (median ± IQR = 0.06 ± 0.03, two-sided sign-rank test; adjusted $P = 2.7 \times 10^{-2}$, $d = 0.45$, $N = 50$). Finally, we did not find any evidence for differential connections between sensory populations that encoded conjunctions to different types of recurrent populations (median ± IQR < 0.006 ± 0.007, two-sided sign-rank test; adjusted $P > 0.39$, $d = 0.10$, $N = 50$).

With regards to recurrent connections, we found stronger connections between excitatory and inhibitory populations than for self-excitation and self-inhibition. Specifically, the average value of naïve weights from excitatory to inhibitory populations and vice versa were significantly larger than the average value of naïve excitatory-excitatory and inhibitory-inhibitory weights (median ± IQR = 0.017 ± 0.009, two-sided sign-rank test; adjusted $P < 10^{-9}$, $d = 0.41$, $N = 50$; Fig. 7b). Among these weights, we found that weights from the inhibitory populations with plastic sensory input (Inh$_{fr}$ and Inh$_{ff}$) to the excitatory populations with plastic sensory input (Exc$_{fr}$ and Exc$_{ff}$) were stronger than the average naïve weights between other excitatory and inhibitory populations. This was reflected in the difference between these average naïve weights being significantly larger than zero

(median ± IQR = 0.036 ± 0.006, two-sided sign-rank test; adjusted $P < 10^{-9}$, $d = 1.31$, $N = 50$). Similarly, the average value of naïve weights from the excitatory populations with no plastic sensory input (Exc$_{rr}$ and Exc$_{rf}$) to the inhibitory population with plastic sensory input and plastic recurrent connections (Inh$_{ff}$) was significantly larger than the average naïve weights between other excitatory and inhibitory recurrent populations (median ± IQR = 0.041 ± 0.005, two-sided sign-rank test; adjusted $P < 10^{-9}$, $d = 1.7$, $N = 50$). Finally, among excitatory populations, those with plastic sensory input had the strongest influence on the output population, as the naïve weights from these populations to the output population was significantly larger than the average value of naïve weights from the excitatory populations with no plastic sensory input to the output population (median ± IQR = 0.015 ± 0.008, two-sided sign-rank test; adjusted $P = 0.02$, $d = 0.51$, $N = 50$; Fig. 7c).

Together, analyses of naïve weights illustrate that learning in environments with a wide range of generalizability results in a specific pattern of connections influencing future learning. Specifically, feature-encoding sensory populations are more strongly connected to inhibitory neurons, whereas object-encoding sensory populations are more strongly connected to excitatory neurons. Moreover, stronger cross connections between inhibitory and excitatory neurons indicate an opponency between the two neural types in shaping complex learning strategies. Although this opponency was expected to account for transition between types of different learning strategies, the ability of excitatory recurrent populations to influence the output population could explain the larger contribution of excitatory populations to the object-based strategy. Specifically, object values can be estimated directly by a single connection from sensory populations to recurrent populations because they do not require integration of information across features and/or conjunctions as is the case for feature-based and conjunction-based strategies. As a result, object values could directly drive excitatory populations which in turn drive the output of the network.

Due to activity dependence of the learning rule, initial stronger connections could be modified more dramatically due to reward feedback in our experiment, and thus, are more crucial for the observed behavior. To directly test this, we used GLMs to fit plastic input weights from sensory units during the course of learning in our experiment. More specifically, we used reward probabilities associated with different aspects of the presented stimulus (features, conjunctions of features, and stimulus identity) to predict flexible connection weights from sensory to recurrent units. This allowed us to measure the rates of change in connection weights due to reward feedback in our task and thus, the contributions of different populations to observed learning behavior. We also measured changes in plastic connections between recurrent populations over time (see Methods section for more details).

We found that connection weights from feature-encoding populations to inhibitory populations showed a value-dependent reduction that was only significant in the connections from sensory populations encoding the informative feature (median ± IQR = −0.34 ± 0.06, two-sided sign-rank test; $P < 0.01$, $d > 0.44$, $N = 50$; Fig. 7d). In contrast, the connection weights from feature-encoding populations to excitatory populations showed a significant value-dependent increase (median ± IQR = 0.47 ± 0.07, two-sided sign-rank test; $P < 0.01$, $d > 0.42$, $N = 50$). At the same time, the connection weights from the sensory populations encoding the informative conjunction to the excitatory populations with plastic sensory input showed only a trend for value-dependent increase (median ± IQR = 0.25 ± 0.07, two-sided sign-rank test; $P = 0.08$, $d = 0.17$, $N = 50$).

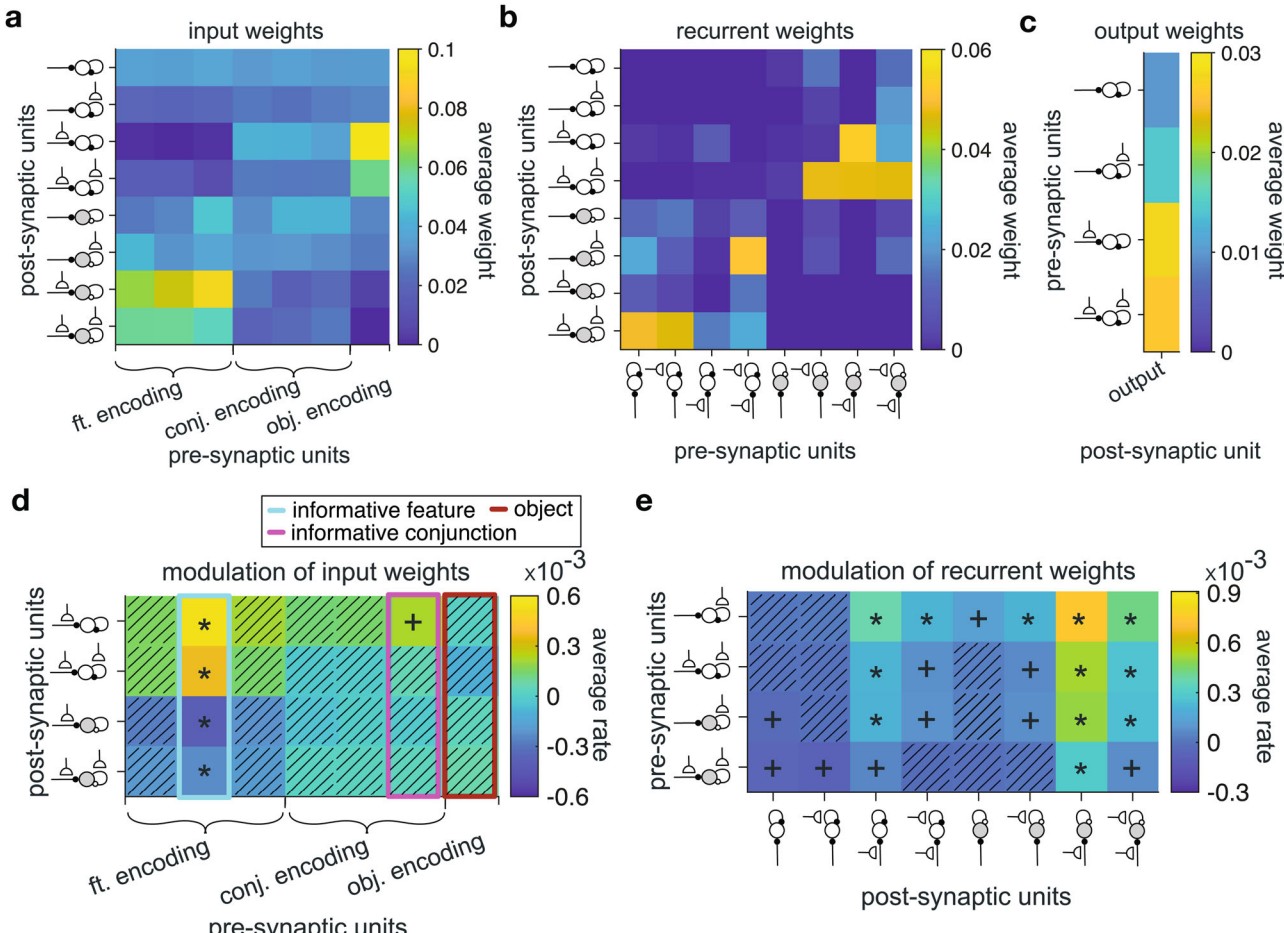

**Fig. 7 RNNs' naïve weights at the end of the training step and their subsequent rates of change due to reward feedback during the simulation of our task. a–c** Plotted is the average strength of the naïve weights from feature-encoding, conjunction-encoding, and object-identity encoding populations to eight types of recurrent populations (indicated by the inset diagrams explained in Fig. 4b) (**a**), naïve weights between eight types of recurrent populations (**b**), and naïve weights from four types of excitatory recurrent populations to the output population (**c**). **d** Plotted is the average rate of value-dependent changes in the connection weights from feature-encoding, conjunction-encoding, and object-identity encoding populations to recurrent populations with plastic sensory input, during the simulation of our task. Asterisks and plus sign indicate two-sided and one-sided significant rates of change (sign-rank test; $P < 0.05$), respectively. Hatched squares indicate connections with rates of change that were not significantly different from zero (two-sided sign-rank test; $P > 0.05$). Highlighted rectangles in cyan, magenta, and red indicate the values for input from sensory units encoding the informative feature, the informative conjunction, and object-identity, respectively. **e** Plot shows the average rates of change in connection weights between recurrent populations. Conventions are the same as in **d**. Source data are provided as a Source Data file.

Finally, analysis of plastic recurrent connections showed that the rates of change in weights to the inhibitory populations with plastic sensory input ($Inh_{rf}$ and $Inh_{ff}$) was modulated more strongly than all other recurrent connections (median ± IQR = $3.5 \times 10^{-4} \pm 1.7 \times 10^{-5}$, two-sided sign-rank test; $P = 3.6 \times 10^{-4}$, $d = 0.33$, $N = 50$; Fig. 7e). Among connections to the these populations, the rates of change in weights from the excitatory populations with plastic recurrent connections ($Exc_{rf}$ and $Exc_{ff}$) was stronger than the rates of change in weights from the inhibitory populations with plastic sensory input ($Inh_{rf}$ and $Inh_{ff}$) (median ± IQR = $2.2 \times 10^{-4} \pm 1.4 \times 10^{-5}$, two-sided sign-rank test; $P = 0.01$, $d = 0.22$, $N = 50$). These results suggest the importance of the inhibitory populations with plastic sensory input for the observed changes in learning strategies over time.

Together, our results on changes in connectivity pattern due to learning in our task illustrate that learning about stimuli leads to simultaneous increase in connection strength from the feature- and conjunction-encoding populations to excitatory populations and a decrease in connection strength from the feature-encoding populations to inhibitory populations, respectively. These simultaneous changes effectively cause excitation and disinhibition of excitatory populations, suggesting a role for a delicate interplay between inhibition and excitation in acquiring and adopting mixed feature- and conjunction-based learning strategies.

These results can be explained by noting that inhibitory populations disinhibit excitatory populations according to a feature-based strategy, and because our learning rules depend on pre- and post-synaptic activity, learning in input connections from sensory populations to recurrent populations become dominated by the feature-based and not the object-based strategy. Because of this value-dependent disinhibition, representations of feature values are reinforced (while suppressing object-based values) in excitatory populations, allowing for the intermediate strategies to drive the output layer. Finally, the absence of changes in the connections from object identity-encoding populations to excitatory populations suggests a role for interaction between excitatory and inhibitory populations in suppressing object-based strategy.

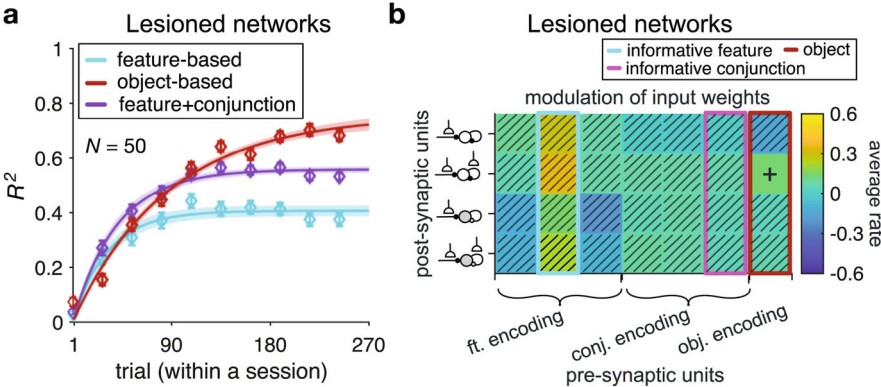

**Fig. 8 Lesioning recurrent connections from the inhibitory populations with plastic sensory input (Inh$_{fr}$ and Inh$_{ff}$) to the excitatory populations with plastic sensory input (Exc$_{fr}$ and Exc$_{ff}$) results in drastic changes in the behavior of the RNNs. a** The plot shows the time course of explained variance ($R^2$) in RNNs' estimates based on different models. Error bars represent s.e.m. The solid line is the average of exponential fits to RNNs' data, and the shaded areas indicate ±s.e.m. of the fit. **b** Plotted is the average rate of value-dependent changes in the connection weights from feature-encoding, conjunction-encoding, and object-identity encoding populations to recurrent populations with plastic sensory input (indicated by the inset diagrams explained in Fig. 4b), during the simulation of our task. The plus sign indicates one-sided significant rate of change (sign-rank test; $P < 0.05$), whereas hatched squares indicate connections with rates of change that were not significantly different from zero (two-sided sign-rank test; $P > 0.05$). Highlighted rectangles in cyan, magenta, and red indicate the values for input from sensory units encoding the informative feature, the informative conjunction, and object-identity, respectively. Source data are provided as a Source Data file.

**Causal role of certain connections in emergence of mixed learning strategies**. As mentioned above, we found strong connections between the inhibitory and excitatory populations with plastic sensory input (Fig. 7b, e). This suggests that these connections could play an important role in the emergence of mixed feature- and conjunction-based learning strategies over time. To test this, in two separate sets of simulations, we lesioned connections from the inhibitory populations with plastic sensory input (Inh$_{fr}$ and Inh$_{ff}$) to the excitatory populations with plastic sensory input (Exc$_{fr}$ and Exc$_{ff}$) and vice versa. We found that RNNs with lesioned connections from the inhibitory populations with plastic sensory input to the excitatory populations with plastic sensory input exhibited a dominant object-based learning strategy (Fig. 8a). Consistent with this result but unlike the intact networks, there was no value-dependent changes in connection weights from the feature- and conjunction-encoding populations to excitatory and inhibitory populations (Fig. 8b). However, we found significant value-dependent changes in connection weights from the object-encoding populations to the recurrent populations with plastic sensory input and plastic recurrent populations (median ± IQR = 0.14 ± 0.03, two-sided sign-rank test; $P = 0.06$, $N = 50$), pointing to the role of excitatory neurons in driving object-based learning.

Consistent with these observations, we also found that in the network with lesioned connections from the inhibitory populations with plastic sensory input to the excitatory populations with plastic sensory input, the difference between the weight of object and the weights of informative feature and informative conjunction was larger than 0 (median ± IQR = 0.06 ± 0.11, two-sided sign-rank test; $P = 4.87 \times 10^{-4}$, $d = 0.34$, $N = 50$; Supplementary Fig. 11). This indicates that the dissimilarity of the stimulus reward probabilities was better predicted by the dissimilarity of response based on objects in these lesioned networks.

In contrast, lesioning the connections from the excitatory populations with plastic sensory input (Exc$_{fr}$ and Exc$_{ff}$) to the inhibitory populations with plastic sensory input (Inh$_{fr}$ and Inh$_{ff}$) did not strongly impact the emergence of mixed feature- and conjunction-based learning (Supplementary Fig. 12a, b). In these lesioned networks, however, only the connection weights from feature-encoding populations to excitatory populations showed a value-dependent increase over time (median ± IQR =

−0.31 ± 0.05, two-sided sign-rank test; $P < 0.01$, $N = 50$, Supplementary Fig. 12c). In addition, we found a decrease in the explanatory power of dissimilarity of the informative feature and the informative conjunction in predicting the dissimilarity of the stimulus reward probabilities compared to the intact network (median ± IQR = −0.19 ± 0.31, two-sided sign-rank test; $P = 2.83 \times 10^{-5}$, $d = 0.40$, $N = 50$; Supplementary Fig. 13). Unlike the intact networks, however, the informative feature and conjunction still better explained the dissimilarity of the stimulus reward probabilities as reflected in the difference between the weights of the informative feature and conjunction and the weight of the objects being larger than 0 (median ± IQR = 0.45 ± 0.27, two-sided sign-rank test; $P < 10^{-8}$, $d = 0.61$, $N = 50$).

Altogether, results of lesioning suggest that the observed mixed feature- and conjunction-based learning strategy mainly relies on recurrent connections from the inhibitory populations with plastic sensory input to the excitatory populations with plastic sensory input as the object-based strategy dominates only in the absence of these specific connections. This happens because once the connections from the inhibitory populations are lesioned, feature-based disinhibition is removed and the object-based strategy dominates. This suggests that the value-based disinhibition of excitatory neurons leads to suppression of object-based strategy and allows for the adoption of intermediate learning strategies. We next tested the importance of learning in these connections using alternative network architectures.

**Plasticity in recurrent connections is crucial for complex learning strategies to emerge**. We trained three alternative architectures of RNNs to further confirm the importance of recurrent connections and reward-dependent plasticity in sensory and recurrent connections. Specifically, we trained RNNs without plastic sensory input, RNNs without plastic recurrent connections, and feedforward neural networks (FFNNs) with only excitatory populations and plastic sensory input.

We found that the RNNs without plastic sensory input were not capable of learning even non-structured stimulus-outcome associations during the training step. Moreover, although the RNNs without plastic recurrent connections and the FFNNs were able to perform the task and learn stimulus-outcome associations,

their behavior was significantly different from the behavior of our human participants. Specifically, the RNNs without plastic recurrent connections only exhibited a dominant object-based learning strategy (Supplementary Fig. 14a). In contrast, the explained variance of fit of FFNNs' estimates over time was best fit by the $F + C_1$ model (Supplementary Fig. 14b), however, the learning time course in this model was different from that of the human participants. Specifically, unlike our experimental results, we did not find any significant difference between the time constant of increase in the weight of the informative feature and the time constant of increase in the weight of the informative conjunction in the FFNN ($\tau_{\text{inf.feature}}$: median $\pm$ IQR $= 37.04 \pm 29.09$, $\tau_{\text{inf.conjunction}}$: median $\pm$ IQR $= 50.18 \pm 30.51$, two-sided sign-rank test; $P = 0.34$, $d = 0.47$, $N = 50$; Supplementary Fig. 14c). Together, these results demonstrate the importance of plasticity in recurrent connections between excitatory and inhibitory populations for the observed emergence of more complex learning strategies and the time constant of their emergence.

## Discussion

Using a combination of experimental and modeling approaches, we investigated the emergence and adoption of multiple learning strategies and their corresponding value representations in more naturalistic settings. We show that in high-dimensional environments, humans estimate reward probabilities associated with each of many stimuli by learning and combining estimates of reward probabilities associated with the informative feature and the informative conjunction. Moreover, we find that feature-based learning is much faster than conjunction-based learning and emerges earlier. These results are not trivial for multiple reasons. First, instead of gradually adopting a combination of feature-based and conjunction-based strategies, participants could simply stop at feature-based learning or gradually transition to object-based learning. This is because in the absence of forgetting (decay in estimates of values over time), an object-based strategy could ultimately provide more accurate estimates of reward probabilities even with less frequent updates. Second, more complex representations of value increase the complexity of learning and decision-making processes and thus, are less desirable.

Analyses of connectivity pattern and response of units in the trained RNNs illustrate that such mixed value representations emerge over time as a distributed code that depends on distinct contributions of inhibitory and excitatory neurons. More specifically, learning about multi-dimensional stimuli results in contrasting connectivity patterns and representations of feature and object values in inhibitory and excitatory neurons, respectively. Through disinhibition, recurrent connections allow gradual emergence of a mixed (feature-based followed by conjunction-based) learning strategy in excitatory neurons. This emergence relies more strongly on connections from inhibitory to excitatory populations because in the absence of these connections, object-based learning can quickly dominate. Moreover, alternative network structures without such connections failed to reproduce the behavior of our human participants, demonstrating the importance of learning in these connections for emergence of conjunction-based learning. Our results thus, provide clear testable predictions about the emergence and neural mechanisms of naturalistic learning.

Our behavioral results support a previously proposed adaptability-precision tradeoff (APT) framework[3,39,40] for understanding competition between different learning strategies. Moreover, they confirm our hypothesis that the complexity of learning strategies depends on the generalizability of the reward environment. Importantly, in the absence of any instruction, human participants were able to detect the level of generalizability of the environment and learn the informative feature and conjunctions of non-informative features. The conjunction-based learning that follows feature-based learning enables the participants to improve accuracy in their learning without significantly compromising the speed, thus improving the APT.

The timescale at which conjunction-based learning emerged was an order of magnitude slower than that of feature-based learning. Although expected, this result provides a critical test for finding the underlying neural architecture. Our results also indicate that humans can learn higher-order associations (conjunctions) when lower-order associations (non-informative features) are not useful. However, this only happens if the environment is stable enough (relative to the timescales of different learning strategies) such that there is sufficient time for slower representations to emerge before reward contingencies change. Ultimately, the timescale of different learning strategies have important implications for learning in naturalistic settings and for identifying their neural substrates[41,42].

The categorization learning and stereotyping literature can provide an alternative but complementary interpretation of our behavioral results[43–46]. A task with generalizable reward schedule can be considered a rule-based reasoning task (i.e., a task in which optimal strategy can be described verbally), whereas a task with a non-generalizable reward schedule can be interpreted as an information integration task (i.e., a task in which information from two or more stimulus components should be integrated). For example, the Competition between Verbal and Implicit Systems (COVIS) model of category learning assumes that rule-based category learning is mediated primarily by an explicit (i.e., hypothesis-testing) system, whereas information integration is dominated by an implicit (i.e., procedural-learning-based) system. According to COVIS, these two learning systems are implemented in different regions of the brain, but the more successful system gradually dominates[47]. In general, feature-based and conjunction-based strategies can be considered as rule-based category learning, whereas object-based strategy can be considered as procedural learning. Thus, in this framework, our results and the APT can be seen as a way of quantifying what factors can cause one learning strategy to dominate the other.

Our training algorithm was designed to allow the network to learn a general solution for learning reward probabilities in multi-dimensional environments. Networks capable of generalizing to new tasks or environments have been the focus of the meta-learning field[48–52] and were used to simulate learning a distribution of tasks[53]. Extending this approach to a learning task with three-dimensional choice options, our modeling results thus suggest that the brain's ability to generalize might arise from principled learning rules along with structured connectivity patterns.

We found distributed representations of reward value in our proposed RNNs. Such representations have been the focus of many studies in memory[54], face- and object-encoding[55–57], and semantic knowledge[58,59], but have not been thoroughly examined in reward learning. Due to similarities between learning and categorization tasks, recurrent connections between striatum and prefrontal cortex have been suggested to play an important role in this process[14,47]. However, future experimental studies are required to investigate the emergence of distributed value representations in the reward learning system.

Based on multiple types of analyses, we show that respectively differential connections of feature-encoding and object-encoding sensory neurons to excitatory and inhibitory populations results in an opponency between representations of feature and object values. This opponency by excitatory and inhibitory neurons

allows for value-based modulation of the excitatory neurons through the inhibitory neurons and enables the adoption of intermediate strategies. Our results can also explain a few existing neural observations. First, we showed that recurrent inputs from certain inhibitory populations in the RNNs result in disinhibition of excitatory populations during the learning. Our RNNs structure does not make any specific assumption on where such disinhibition is originating from, and therefore it can be further extended to explain the effects of disinhibition in subcortical areas, such as the basolateral amygdala[60–63] or striatum[64,65], on associative learning. Second, our findings could explain a previously reported link between disruption in recurrent connections between excitatory and inhibitory neurons in basal ganglia and deficits in the adoption of proper learning strategies in Parkinson's Disease[66–69]. Third, we observed a higher degree of similarity between activity of inhibitory recurrent populations and the informative feature value, which is in line with recent findings on the prevalence of feature-specific error signal in narrow-spiking neurons[11]. Finally, our results predict that error signals in neurons with plastic sensory input are skewed toward conjunction-based learning strategies.

Our experimental paradigm and computational modeling have a few limitations. First, because of the difficulty of the task, we used a stable, multi-dimensional reward environment. However, humans and animals are often required to learn reward and/or predictive values in changing environments. Future experiments and modeling are required to explore learning in dynamic multi-dimensional environments. Second, recent studies have suggested that reward can enhance processing of behaviorally relevant stimuli by influencing different low-level and high-level processes such as sensory representation, perceptual learning, and attention[11,70–77]. Our RNNs only incorporate low-level synaptic plasticity and thus, future studies are needed to understand the contribution of high-level processes (such as attention) in naturalistic value-based learning. Finally, we utilized a backpropagation algorithm to train the RNNs, which often is viewed as non-biological. Our aim was to offer a general framework to test hypotheses about learning in naturalistic environments. Nonetheless, recent studies have proposed dendritic segregation and local interneuron circuitry as a possible mechanism that can approximate this learning method[78–82]. Future progress in mapping the backpropagation method to learning in cortical structures would clarify the plausibility of this training approach.

Together, our study provides insight into both why and how more complex learning strategies emerge over time. On the one hand, we show that mixed learning strategies are adopted because they can provide an efficient intermediate learning strategy that is still much faster than learning about individual stimuli or options but is also more precise than feature-based learning alone. On the other hand, we provide clear testable predictions about neural mechanisms underlying naturalistic learning and thus, address both representations and functions of different neural types and connections.

## Methods

**Participants**. Participants were recruited from the Dartmouth College student population (ages 18–22 years). In total, 92 participants were recruited (66 females) and performed the experiment. We excluded participants whose performance was not significantly different from chance (0.5) indicating that they did not learn the task. To that end, we used a performance threshold of 0.55, equal to 0.5 plus two times s.e.m., based on the average of 400 trials after excluding the first 32 trials of the experiment. This resulted in the exclusion of 25 participants from our dataset. We also analyzed the behavior of excluded participants to show that indeed, they did not learn the task (Supplementary Fig. 4). No participant had a history of neurological or psychiatric illness. Participants were recruited through the Department of Psychological and Brain Sciences experiment scheduling system at Dartmouth College. They were compensated with a combination of money and T-points, which are extra-credit points for classes within the Department of

Psychological and Brain Sciences at Dartmouth College. The base rate for compensation was $10/hour or 1 T-point/hour. Participants were then additionally rewarded based on their performance by up to $10/hour. All experimental procedures were approved by the Dartmouth College Institutional Review Board, and informed consent was obtained from all participants before the experiment.

**Experimental paradigm**. To study how appropriate value representations and learning strategies are formed and evolve over time, we designed a probabilistic learning paradigm in which participants learned about visual stimuli with three distinct features (shape, pattern, and color) by choosing between pairs of stimuli followed by reward feedback (choice trials), and moreover, reported their learning about those stimuli in separate sets of estimation trials throughout the experiment (Fig. 1). More specifically, participants completed one session consisting of 432 choice trials (Fig. 1a) interleaved with five bouts of estimation trials that occurred after choice trial 86, 173, 259, 346, and 432 (Fig. 1a bottom inset). During each choice trial, participants were presented with a pair of stimuli and were asked to choose the stimulus that they believed would provide the most reward. These two stimuli were drawn pseudo-randomly from a set of 27 stimuli, which were constructed using combinations of three distinct shapes, three distinct patterns, and three distinct colors. The two stimuli presented in each trial always differed in all three features. Selection of a given stimulus was rewarded (independently of the other stimulus) based on a reward schedule (set of reward probabilities) with a moderate level of generalizability. This means that reward probability associated with some but not all stimuli could be estimated by combining the reward probabilities associated with their features (see Eq. 1 below). More specifically, only one feature (shape or pattern) was informative about reward probability whereas the other two were not informative. Although the two non-informative features were on average not predictive of reward, specific combinations or conjunctions of these two features were partially informative of reward (Fig. 1b). Finally, during each estimation trial, participants were asked to provide an estimate about the reward probability associated with a given stimulus (Fig. 1a bottom inset).

**Reward schedule**. The reward schedule (e.g., Fig. 1b) was constructed to test the adoption of different learning strategies. To that end, we considered three types of learners with distinct strategies for estimating reward and/or predictive value of individual stimuli. Assume that stimuli or objects (O) have $m$ features (e.g., color, pattern, and shape), each of which can take $n$ instances (e.g., yellow, solid, and triangles), indicated as $F_{i,j}$ for the feature instance $j$ of feature $i$, where $i = \{1, ..., m\}$ and $j = \{1, ..., n\}$. In contrast to an object-based learner that directly estimates reward probability for each stimulus using reward feedback, a feature-based learner uses the average reward probability for each feature instance to estimate the reward probability associated with each stimulus in two steps. First, the average reward probability for a given feature instance (e.g., color yellow) can be computed by averaging the reward probability of all stimuli that contain that feature instance (e.g., all yellow stimuli); $\bar{p}_r(F_{i,j}) = (1/n^{m-1}) \sum_{O_a \text{contains} F_{ij}} p_r(O_a)$ or by multiplying the likelihood ratios of all stimuli that contain that feature instance: $\bar{p}_r(F_{i,j}) = (\prod_{O_a \text{contains} F_{ij}} LL_r(O_a))^{1/n^{m-1}}$. Second, reward probability for a stimulus $O_a$, $\tilde{p}_r(O_a)$, can be estimated by combining the reward probability of features of that stimulus using Bayes theorem:

$$\tilde{p}_r(O_a) = (\overline{p_r}(F_{1,j}) \times \overline{p_r}(F_{2,k}) \times \dots) / (\overline{p_r}(F_{1,j}) \times \overline{p_r}(F_{2,k}) \times \dots + (1 - \overline{p_r}(F_{1,j}))$$
$$\times (1 - \overline{p_r}(F_{2,k})) \times \dots) \text{ for } O_a \text{ containing } F_{1,j}, F_{2,k}, \text{etc} \quad (1)$$

These estimated reward probabilities constitute the estimated reward matrix based on features. The rank order of probabilities in the estimated reward matrix, which determines preference between stimuli, is similar to that of the fully generalizable reward matrix whereas the exact probabilities may differ slightly. Note that although we assumed an optimal combination of feature values (using Bayes theorem), a more heuristic combination of feature values results in qualitatively similar results[3,76].

Similarly, a mixed feature- and conjunction-based learner combines the reward probability for one or more feature instances, $F_{i,j}$ (where $i = I \subseteq \{1, ..., m\}$ and $j = \{1, ..., n\}$), and the conjunctions of the other remaining features (e.g., solid triangle, indicated as $C_{l,k}$ for the conjunction instance $k$, $k = \{1, ..., n^{m-|I|}\}$ of conjunction type $l$, where $l_i = \{1, ..., m\}$-I) to estimate reward probabilities of stimuli in three steps. First, the average reward probability associated with one or more feature instances can be calculated as above. Second, the average reward probability associated with one or more conjunctions of remaining features can be computed by averaging the reward probabilities of all stimuli that contain that conjunction instance (e.g., all solid triangle stimuli); $\tilde{p}_r(C_{k,l}) = (1/n^{m-|I|}) \sum_{O_a \text{contains} C_{k,l}} p_r(O_a)$, or by multiplying the likelihood ratios of all stimuli that contain that conjunction instance; $\bar{p}_r(C_{k,l}) = (\prod_{O_a \text{contains} C_{k,l}} LL_r(O_a))^{1/n^{m-|I|}}$. Finally, reward probability for a stimulus $O_a$, $\tilde{p}_r(O_a)$, can be estimated by combining the

reward probabilities of features and conjunctions using the Bayes theorem:

$$\tilde{p}_r(O_a) = (\overline{p_r}(F_{1,j}) \times \overline{p_r}(C_{23,k}) \times \dots )(\overline{p_r}(F_{1,j}) \times \overline{p_r}(C_{23,k}) \times \dots$$
$$+ (1 - \overline{p_r}(F_{1,j})) \times (1 - \overline{p_r}(C_{23,k})) \times \dots ) \text{ for } O_a \text{ containing } F_{1,j}, C_{23,k}, \text{ etc.}$$
$$(2)$$

**Generalizability index.** To define generalizability indices, we used the Spearman correlation between the stimuli's actual reward probability and estimated reward probability based on their individual features or between stimuli's actual reward probability and estimated reward probability based on the mixture of the informative feature and the conjunctions of the two non-informative features (see previous section). Based on this definition, the generalizability index can take on any values between −1 and 1.

**Using estimates of reward probabilities to assess learning strategies.** We utilized estimates of reward probabilities provided by both the human participants (during estimation trials) and those extracted from the trained RNNs to examine how these estimates were constructed and determine the underlying learning strategies. Specifically, first, we used GLMs to predict both participants' and RNNs' estimates of reward probabilities as a function of each of the following variables: actual reward probabilities assigned to each stimulus (object-based term); reward probability estimates based on the combination of the reward probabilities associated with individual features (feature-based term; Eq. 1); reward probability estimates based on the combination of the reward probability of the informative feature and the conjunctions of the other two non-informative features (mixed feature- and conjunction-based term; Eq. 2); and an additional constant term to capture the overall bias. Second, we used a single stepwise GLM to predict participants' estimates of reward probabilities using the aforementioned object-based and feature-based terms as well as those based on the reward probability of the conjunctions of the two non-informative features.

**Fitting the time course of regression weights to predict participants' estimates.** To quantify the learning time course using participants' estimates of reward probabilities, in addition to $R^2$ of the GLMs, we also fit stepwise regression normalized weights for the informative feature and conjunction, and object (stimulus identity) using an exponential function:

$$y(t) = y_{ss} - (y_{ss} - y_0)\exp\left(\frac{-t}{\tau}\right) \quad (3)$$

where $y_0$ and $y_{SS}$ are the initial and steady-state values of weights, $\tau$ is the time constant for approaching steady state, and $t$ represents the trial number in a session.

**Testing behavioral predictions of learning models.** To directly assess the effects of learning strategy on choice behavior, we defined a few quantities we refer to as differential response separately for the informative and non-informative features, and conjunctions of non-informative features. Differential response for individual features is defined as the difference between two conditional probabilities: (1) conditional probability of selecting stimuli that contain only one of the two features (informative or non-informative) of the stimulus selected and rewarded in the previous trial when these stimuli were paired with a stimulus that did not share any feature with the previously rewarded stimulus; and (2) a similar conditional probability when the previous trial was not rewarded (see Supplementary Fig. 5). Similarly, we defined differential response for conjunctions of non-informative features by considering stimuli that contain the conjunction of non-informative features of the stimulus selected in the previous trial.

**Model fitting.** To capture participants' learning and choice behavior on a trial-by-trial basis, we used 24 different reinforcement learning (RL) models based on feature-based, mixed feature- and conjunction-based, and object-based strategies (see below). These models were fit to choices from individual participants by minimizing the negative log likelihood (LL) of the predicted choice probability given different model parameters using the fminsearch function in MATLAB (Mathworks). We computed three measures of goodness-of-fit in order to determine the best model: average negative log likelihood, Akaike information criterion (AIC), and Bayesian information criterion (BIC). In addition, to compare the ability of different models in fitting choice behavior over time, we used AIC and BIC per trial[10,34], denoted as $\text{AIC}_p$ and $\text{BIC}_p$:

$$\text{AIC}_p(t) = -2\text{LL}(t) + 2k/N_{\text{trials}} \quad (4)$$

$$\text{BIC}_p(t) = -2\text{LL}(t) + 2k\log(N_{\text{trials}})/N_{\text{trials}} \quad (5)$$

where $k$ indicates the number of parameters in a given model, $t$ represents the trial number, $\text{LL}(t)$ is the log likelihood in trial $t$, and $N_{\text{trials}}$ is the number of trials in the experiment. By dividing the penalty terms in AIC and BIC by the number of trials, we ensure that the sum of $\text{AIC}p(t)$ and $\text{BIC}p(t)$ over all trials is equal to AIC and BIC, respectively. The smaller values for these measures indicate a better fit of choice behavior.

**Object-based models.** In this group of models, the reward probability associated with each stimulus is directly estimated from reward feedback in each trial using a standard RL model. For example, in the chosen-update object-based RL, only the reward probability associated with the chosen stimulus is updated in each trial. This update is done via separate learning rates for rewarded or unrewarded trials using the following equations, respectively:

$$V_{\text{choS}}(t + 1) = V_{\text{choS}}(t) + \alpha_{\text{rew}}\left(1 - V_{\text{choS}}(t)\right), \text{ if } r(t) = 1$$

$$V_{\text{choS}}(t + 1) = V_{\text{choS}}(t) - \alpha_{\text{unr}}\left(V_{\text{choS}}(t)\right), \text{ if } r(t) = 0 \quad (6)$$

where $t$ represents the trial number, $V_{\text{choO}}$ is the estimated reward probability associated with the chosen stimulus, $r(t)$ is the reward outcome on the chosen stimulus (1 for rewarded, 0 for unrewarded), and $\alpha_{\text{rew}}$ and $\alpha_{\text{unr}}$ are the learning rates for rewarded and unrewarded trials. The reward probability associated with the unchosen stimulus is not updated in this model.

In the full-update object-based RL, the reward probabilities associated with both stimuli presented in a given trial are updated. Specifically, while reward probability associated with the chosen stimulus is updated based on Eq. 6, the value of unchosen stimulus is also updated based on reward feedback on that stimulus according to the following equations:

$$V_{\text{uncS}}(t + 1) = V_{\text{uncS}}(t) + \alpha_{\text{rew}}\left(1 - V_{\text{uncS}}(t)\right), \text{ if } r_{\text{unc}}(t) = 1$$

$$V_{\text{uncS}}(t + 1) = V_{\text{uncS}}(t) - \alpha_{\text{unr}}\left(V_{\text{uncS}}(t)\right), \text{ if } r_{\text{unc}}(t) = 0 \quad (7)$$

where $t$ represents the trial number, $V_{\text{uncS}}$ is the estimated reward probability associated with the unchosen stimulus, and $r_{\text{unc}}(t)$ is the reward outcome on the unchosen stimulus.

The estimated reward probability values are then used to compute the probability of selecting between the two stimuli in a given trial (S1 and S2) based on a logistic function:

$$\text{logit } P_{S1}(t) = w_o\left(V_{S1}(t) - V_{S2}(t)\right) + bias \quad (8)$$

where $P_{S1}$ is the probability of choosing stimulus 1, $V_{S1}$ and $V_{S2}$ are the estimated reward probability associated with stimuli 1 and 2, respectively, bias measures a response bias toward the left or right stimulus to capture the participant's location bias, and $w_o$ determines the influence of difference in reward probabilities associated with a pair of presented stimuli (objects) on choice.

**Feature-based models.** In this group of models, the reward probability associated with each stimulus is computed by combining the reward probability associated with features of that stimulus, which are estimated from reward feedback using a standard RL model. The update rules for the feature-based RL models are identical to the object-based ones (chosen-update and full-update models), except that the reward probability associated with the chosen (unchosen) stimulus is replaced by the reward probability associated with features of the chosen (unchosen) stimulus.

Similar to object-based RL models, the probability of choosing a stimulus is determined based on the logistic function of the difference between the estimated reward probability associated with the two presented stimuli in a given trial:

$$\text{logit } P_{S1}(t) = \sum_{Fi=\{shape, pattern, color\}} w_{Fi}\left(V_{Fi(S1)}(t) - V_{Fi(S2)}(t)\right) + bias \quad (9)$$

where $V_{Fi(S1)}$ and $V_{Fi(S2)}$ are the reward probabilities associated with a given feature (out of three possible features) of stimuli 1 and 2, respectively, bias measures a response bias toward the left stimulus to capture the participant's location bias, and three values of $\mathbf{w}_F$ determine the influence of difference in reward probabilities associated with feature $F$ of presented stimuli on choice.

**Mixed feature- and conjunction-based models.** In this group of models (referred to as the F+C models for abbreviation), the reward probability associated with each stimulus is computed by combining the reward probabilities associated with one feature and the conjunction of the other two features of that stimulus, all of which are estimated from reward feedback using a standard RL model. In these models, we assume that participants identify one feature as the likely informative feature early in the experiment and subsequently learn about the conjunctions of the other two features. The update rules for the mixed feature- and conjunction-based RL models are identical to the previous models, except that the reward probability associated with the chosen or unchosen stimulus is replaced by the reward probability associated with one of the three features (which we refer to as the learned feature) or the conjunctions of the other two features (learned conjunction) of the chosen or unchosen stimulus. Additionally, we considered separate learning rates for rewarded and unrewarded trials ($\alpha_{\text{rew}}, \alpha_{\text{unr}}$) when updating reward probabilities associated with the learned feature and conjunction. This setting results in three different mixed feature- and conjunction-based models depending on the learned feature and conjunction. In the F+C$_1$ model, we assume that the informative feature and the conjunction of the other two non-informative features are being learned; for example, values of shapes and conjunctions of colors and patterns are learned (Fig. 1). In the F+C$_2$ and F+C$_3$ models, one of the two non-informative features is learned along with the conjunction of the other non-informative feature and the informative feature. In the example reward schedule shown in Fig. 1,

reward probabilities associated with colors and conjunctions of shapes and patterns are learned in F+C$_2$, whereas reward probabilities associated with patterns and conjunctions of shapes and colors are learned in F+C$_3$.

Finally, the probability of choosing a stimulus is determined based on the logistic function of the difference between the total estimated reward probabilities associated with each of the two presented stimuli in a given trial:

$$\text{logit } P_{S1}(t) = w_{\text{F}}\left(V_{\text{F(S1)}}(t) - V_{\text{F(S2)}}(t)\right) + w_{\text{C}}\left(V_{\text{C(S1)}}(t) - V_{\text{C(S2)}}(t)\right) + bias$$
(10)

where $V_{\text{F(S1)}}$ and $V_{\text{F(S2)}}$ are reward probabilities associated with a feature of stimuli 1 and 2, respectively, $V_{\text{C(S1)}}$ and $V_{\text{C(S2)}}$ are the reward probabilities associated with the conjunctions of the other two features of stimuli 1 and 2, respectively, bias measures a response bias toward the left stimulus to capture the participant's location bias, and $w_{\text{F}}$ and $w_{\text{C}}$ determine the influence of the difference in reward probabilities associated with the learned feature and learned conjunction of presented stimuli on choice, respectively.

**Mixed feature- and object-based models.** In this group of models (referred to as F+O models for abbreviation), the reward probability associated with each stimulus is computed by combining the reward probabilities associated with one of the three features (learned feature) and the reward probability associated with stimulus estimated using reward feedback. In these models, we assume that participants identify one feature as the likely informative feature early in the experiment and combine this learning with what they learn about individual stimuli later in the experiment. The update rules for the mixed feature- and object-based RL models are similar to the previous models. Additionally, we considered separate learning rates for rewarded and unrewarded trials ($\alpha_{\text{rew}}$, $\alpha_{\text{unr}}$) when updating reward probabilities associated with the learned feature and each stimulus. This setting results in three different mixed feature- and object-based models depending the learned feature. In the F$_1$+O model, we assume that the informative feature and the objects are being learned; for example, values of shapes and objects are learned (Fig. 1). In the F$_2$+O and F$_3$+O models, one of the two non-informative features is learned along with the value of the objects. In the example reward schedule shown in Fig. 1, reward probabilities associated with colors and objects are learned in F$_2$+O, whereas reward probabilities associated with patterns and objects are learned in F$_3$+O.

Finally, the probability of choosing a stimulus is determined based on the logistic function of the difference between the total estimated reward probabilities associated with each of the two presented stimuli in a given trial:

$$\text{logit } P_{S1}(t) = w_{\text{F}}\left(V_{\text{F(S1)}}(t) - V_{\text{F(S2)}}(t)\right) + w_{\text{O}}\left(V_{S1}(t) - V_{S2}(t)\right) + bias$$
(11)

where $V_{\text{F(S1)}}$ and $V_{\text{F(S2)}}$ are reward probabilities associated with a feature of stimuli 1 and 2, respectively, $V_{S1}$ and $V_{S2}$ are the reward probabilities associated with the stimuli 1 and 2, respectively, bias measures a response bias toward the left stimulus to capture the participant's location bias, and $w_{\text{F}}$ and $w_{\text{O}}$ respectively determine the influence of the difference in reward probabilities associated with the learned feature of presented stimuli and objects on choice.

**RL models with decay.** Additionally, we investigated the effect of forgetting reward probabilities associated with the unchosen objects, conjunctions of features, or feature(s) in the chosen-update models by introducing a decay in estimated probabilities that has been shown to capture some aspects of learning, especially in multi-dimensional tasks[3,6]. More specifically, reward probabilities associated with unchosen stimuli, conjunctions of features, or feature(s) decay to 0.5 with a rate of $d$, as follows:

$$V(t+1) = V(t) - d \times (V(t) - 0.5)$$
(12)

where $t$ represents the trial number and $V$ is the estimated reward probability associated with a stimulus, or a conjunction of two features, or a feature.

**Recurrent neural network.** To understand computations and neural mechanisms underlying the emergence of different learning strategies, we used recent methods for training recurrent neural networks[24–28,30,31] to construct biologically inspired recurrent networks of point neurons endowed with plausible, reward-dependent Hebbian learning rules[35–37]. Recurrent design was chosen to ensure that the network can demonstrate long-term complex dynamics when learning using reward feedback. We first used stochastic gradient descent (SGD) to train recurrent neural networks (RNNs) consisting of excitatory and inhibitory units to learn input-output associations for estimating reward probabilities and then used the trained RNNs to perform our task. The first training, which was done in a series of multi-dimensional environments (see below), was to allow the networks to learn a general task of learning and estimating reward probabilities. Without such training, these networks would not have the connectivity pattern necessary to choose between multi-dimensional stimuli.

In the first (training) step, we trained RNNs to learn the temporal dynamics of input-output associations and a set of reward probabilities randomly associated with 27 stimuli used in our task. This was done to enable the network to learn a

universal solution for learning and estimating reward probabilities in environments with different levels of generalizability. We used SGD to train the network to learn reward probabilities using a three-factor, reward-dependent Hebbian learning rule for weights from sensory units to recurrent units and between recurrent units. Reward probabilities were drawn from a uniformly random distribution between 0 and 1 for each training session that consisted of 270 trials. Every 50 iterations, RNNs' performance (equal to the mean squared error of reward probability estimates at the end of the session) in a task with reward probabilities similar to our experimental paradigm was calculated. The RNNs were trained until this performance matched the average performance of our participants. In the second step (simulation of the experiment), we stopped SGD and simulated the behavior of the trained RNNs in a session with reward probabilities similar to our experimental paradigm and compared the behavior of the trained RNNs with that of our participants. In this step, only plastic connections were modulated after each reward feedback.

The overall structure of the simulated task was similar to our experimental paradigm with a few exceptions. Specifically, only one stimulus was shown in each trial and the network had to learn the reward probability associated with that stimulus. Moreover, in each trial, networks received visual information about the presented stimulus (i.e., distinct sensory inputs representing features, conjunction of features, and object-identity) as a tonic input and had to learn reward probability associated with that stimulus using reward feedback provided in each trial. We did not require our networks to demonstrate working memory or make decisions between stimuli to be able to better isolate the necessary structure for multi-dimensional reward learning. Using this approach, we avoided complexity related to decision-making processes and mainly focused on learning aspects of our experimental paradigm. Nonetheless, we also simulated choice data generated by the trained RNNs. Specifically, without retraining the networks, we added a decision layer after the output layer to generate binary choice between pairs of stimuli (using a logistic function) and fit simulated choice data (similar to our participants) to show our results and main conclusion do not depend on the choice mechanism.

**Network structure.** Networks consist of three layers with different types of units mimicking different populations of neurons: 63 sensory units, 120 recurrent units, and one output unit (Fig. 4). Recurrent units receive a set of $N_{\text{in}} = 63$ time-varying inputs ($\mathbf{u}_t$) from the sensory units and were trained to produce an output $z_t$. Inputs encode stimuli-relevant sensory information, and the output represented the estimate of the network for reward probability. Input to the recurrent units include the output of 9 nodes representing all features, 27 nodes representing conjunctions of features, and 27 nodes representing object identity of the stimuli. The activity of recurrent units follows the continuous dynamical equation as below (bold capital letters refer to matrixes, bold small letters refer to vectors and italic letters refer to scalars):

$$\tau\dot{\mathbf{x}} = -\mathbf{x} + \mathbf{W}^{\text{rec}}\mathbf{r} + \mathbf{W}^{\text{in}}\mathbf{u} + \mathbf{b}^{\text{rec}} + \sqrt{2\tau\sigma_{\text{rec}}^2}\boldsymbol{\zeta}$$
(13)

$$\mathbf{r} = [\mathbf{x}]_+$$

where $\tau$ is the time constant of the activity in the recurrent units, $\mathbf{W}^{\text{in}}$ is an $N_{\text{in}} \times N$ matrix of connection weights from the sensory units to recurrent units, $\mathbf{W}^{\text{rec}}$ is an $N \times N$ matrix of recurrent connection weights, $\mathbf{b}^{\text{rec}}$ is a bias term, $\boldsymbol{\zeta}$ is Gaussian white noise processes with zero mean and unit variance, $\sigma_{\text{rec}}$ is the strength of the intrinsic noise to the network, and $[]_+$ is the rectifying linear function (ReLu). The output unit $z$ reads out linearly from the network as below:

$$z = \mathbf{w}^{\text{out}}\mathbf{r}$$
(14)

where $\mathbf{w}^{\text{out}}$ is an $1 \times N$ vector of connection weights from the recurrent units to the output unit. To solve the resulting dynamical system, we used the first-order Euler approximation with a time-discretization step $\Delta t$ to arrive at the following equations:

$$\mathbf{x}_t = (1-\alpha)\mathbf{x}_{t-1} + \alpha(\mathbf{W}_{t-1}^{\text{rec}}\mathbf{r}_{t-1} + \mathbf{W}_{t-1}^{\text{in}}\mathbf{u}_t + \mathbf{b}^{\text{rec}} + \sqrt{2\alpha^{-1}\sigma_{\text{rec}}^2}\boldsymbol{\zeta})$$
(15)

$$\mathbf{r}_t = [\mathbf{x}_t]_+$$

$$z_t = \mathbf{w}^{\text{out}}\mathbf{r}_t$$

where $\alpha = \Delta t/\tau$ and $t = [0, T]$ is the time elapsed within a single trial. The network received noisy sensory input from 63 populations of sensory encoding units ($\mathbf{u}_t$) as follows:

$$\mathbf{u}_t = \mathbf{s} \times (\mathbb{H}_{t-0.5} - \mathbb{H}_{t-1.5}) + \sqrt{2\tau\sigma_{\text{in}}^2}\boldsymbol{\zeta}$$
(16)

where $\mathbb{H}$ is the Heaviside function allowing stimuli onset at $0.5s$ and stimuli offset at $1.5s$, $\mathbf{s}$ is a vector 0's and 1's with 1's corresponding to feature, conjunctions of features, and stimulus identity of the presented stimulus, and $\sigma_{\text{in}}$ is the strength of the input noise. In our simulations, we set $\Delta t = 0.02s$, $\tau = 0.1s$, $\sigma_{\text{rec}} = 0.01$, and $\sigma_{\text{in}} = 0.01$.

Critically, we designed our network to obey some of biological constraints observed in mammalian cortex. First, we constrained populations to have purely

excitatory or inhibitory effects and hypothesized that excitatory populations (Exc) outnumber inhibitory populations (Inh) by a ratio of 4 to 1. Second, we assumed that all inputs and outputs of the network were long-range excitatory inputs from an upstream circuit, and therefore all elements of $\mathbf{W}^{in}$ and $\mathbf{w}^{out}$ were forced to be non-negative. Finally, to impose excitatory or inhibitory effects on recurrent connections, we implemented $\mathbf{W}_t^{rec}$ as the product between a matrix for which all entries were trained and forced to be non-negative ($\mathbf{W}_t^{rec}$) and a fixed diagonal matrix with 1 s corresponding to excitatory populations and $-1$ s corresponding to inhibitory populations on its main diagonal[25,27].

To enable the networks to learn a universal structure of our task, we embedded three-factor reward-dependent Hebbian learning in selected weights of the RNNs[35]. Specifically, we hypothesized that a combination of pre- and post-synaptic activity ($F(\text{pre}_i, \text{post}_j)$) and the modulator signal ($D_t$) results in an update of only the recurrent and input weights ($\mathbf{W}_t^{rec}$, and $\mathbf{W}_t^{in}$) as follows:

$$\tau_W \frac{d}{dt} \mathbf{W}_t = \boldsymbol{\omega}_t - \mathbf{W}_t \tag{17}$$

$$\frac{d}{dt} \boldsymbol{\omega}_t \left( \text{pre}_i, \text{post}_j \right) = D_t \times F \left( \text{pre}_i, \text{post}_j \right)$$

$$F \left( \text{pre}_i, \text{post}_j \right) = c_{pre} \bar{r}_t^i + c_{post} \bar{r}_t^j + c_{corr} \bar{r}_t^i \bar{r}_t^j$$

where $c_{pre}, c_{post}, c_{corr}$ are the pre-, post-synaptic, and the correlation terms, $\bar{r}$ represents the firing rate that is lowpass filtered with time constant of $0.1s$, and $\tau_W = 0.1s$ is the time constant of the weights' update. The modulator signal $D_t$ was defined as follows:

$$D_t = \begin{cases} R_t, & t \in T_R \\ N/A, & o.w. \end{cases} \tag{18}$$

where $T_R = [1.75s, 2s]$ represents the time when reward feedback is present, and $R_t$ denotes the binary reward (+1 when reward was present and $-1$ when reward was absent).

We assigned each inhibitory or excitatory population to one of four types of populations: (a) $\text{Exc}_{rr}$ and $\text{Inh}_{rr}$ corresponding to populations with no plastic sensory or recurrent connections (rigid weights indicated by subscript r); (b) $\text{Exc}_{fr}$ and $\text{Inh}_{fr}$ corresponding to populations with plastic sensory input only (flexible weights indicated by subscript f); (c) $\text{Exc}_{rf}$ and $\text{Inh}_{rf}$ corresponding to populations with plastic recurrent connections only; and (d) $\text{Exc}_{ff}$ and $\text{Inh}_{ff}$ corresponding to populations with plastic sensory input and plastic recurrent connections. These eight populations cover all combinations of possible rigid and flexible input and recurrent connections in excitatory and inhibitory populations.

Finally, we simulated three alternative versions of our model. We trained RNNs without plastic sensory input, RNNs without plastic recurrent connections, and feedforward neural network (FFNNs) with only excitatory populations with plastic sensory input.

**RNN training procedure**. In the first training step, we used the Adam version of SGD[83] to train the networks. The objective function $\epsilon$ to be minimized was computed by time-averaging the squared errors between the network output $z_t$ and the target output $z_t'$ ($\mathcal{L}$) in addition to using the $L_2$-norm regularization term ($R$) for encouraging sparse weights/activation patterns. The objective function was calculated as follows:

$$\epsilon = \frac{1}{N_{trials}} \sum_{n=1}^{N_{trials}} (\mathcal{L}_n + \lambda_r R_n) \tag{19}$$

$$\mathcal{L}_n = \frac{1}{N_{time}} \sum_{t=1}^{N_{time}} (z_t - z_t')^2$$

$$R_n = \frac{1}{NN_{time}} \sum_{i=1}^{N} \sum_{t=1}^{N_{time}} (\mathbf{r}_t)_i^2$$

where $\lambda_r = 0.01$ determines the effect of the regularization term $R_n$. The default set of parameters were used for the training where the learning rate was set to 0.001 and the decay rate for the first- and second-moment estimates were set to 0.9 and 0.999, respectively. Model parameters including the learning and decay rates as well as the regularization term were determined by a coarse grid search to achieve a faster training. During training, we adjusted $\mathbf{W}_0^{in}, \mathbf{W}_0^{rec}, \mathbf{w}^{out}, \mathbf{b}^{rec}, \mathbf{b}^{out}, \mathbf{x}_0, (c_{pre}, c_{post}, c_{corr})_{U \rightarrow E/I}$, and $(c_{pre}, c_{post}, c_{corr})_{E/I \rightarrow E/I}$ parameters, where $\mathbf{W}_0^{in}$ and $\mathbf{W}_0^{rec}$ are the initial connection weights and $\mathbf{x}_0$ refers to the initial neuronal activity at time $t = 0$ in each trial. The network was trained with randomly generated reward probabilities between 0 and 1 for each session where target output $z_t'$ was defined as the reward probability of the stimuli during the choice period ($T_{ch} = [1s, 1.5s]$) and zero activity before the stimuli onset ($[0s, 0.5s]$). Each training session consisted of $N_{trials} = 270$ trials of $T = 2s$, equivalent to a session in our experimental paradigm. We used mean squared error of reward probability estimates at the end of a session (with reward probabilities equal to our experimental paradigm) to define network performance and to stop training. Specifically,

training was continued until each network achieved a final estimation error similar to that observed in participants' last reported estimates.

**Simulating the experiment with the trained RNNs**. In the second step, we stopped SGD and simulated the behavior of the trained RNNs in a session with reward probabilities equal to our experimental paradigm. Specifically, during these simulations $\mathbf{W}_0^{in}, \mathbf{W}_0^{rec}, \mathbf{w}^{out}, \mathbf{b}^{rec}, \mathbf{b}^{out}, \mathbf{x}_0, (c_{pre}, c_{post}, c_{corr})_{U \rightarrow E/I}$, and $(c_{pre}, c_{post}, c_{corr})_{E/I \rightarrow E/I}$ parameters were fixed at the values found in the training step while plastic connections were modulated after each reward feedback according to Eqs. 17–18.

**Adjustment for multiple comparisons**. Using the described formulations, we initially trained and analyzed $N = 20$ RNNs. According to the results obtained from these networks, we trained and analyzed a new set of $N = 50$ RNNs. As a result, for the comparison of naïve weights ($\mathbf{W}_0^{in}, \mathbf{W}_0^{rec}, \mathbf{w}^{out}$), we adjusted the $P$-values according to the number of comparisons we performed.

**Analysis of response in recurrent units**. To explore the dynamics of population activity in our networks, we applied principal component analyses (PCA) to the response of excitatory recurrent units of the trained RNNs because this response determines the output of the networks. More specifically, we performed three separate PCAs on the activity of excitatory recurrent units during the simulated experiment (repeated 100 times to obtain smoother results). This includes: PCA on the response of excitatory recurrent units to all stimuli at the beginning of each session before the network has learned about the reward environment; PCA on the response of excitatory recurrent units to all stimuli at the end of each session when the network has fully learned the task; and PCA on the response of excitatory recurrent units to all stimuli during the choice period throughout each session.

Additionally, to identify value representations associated with different strategies in our network model, we examined the dissimilarity in the response of recurrent units to all stimuli and its relationship to dissimilarity of reward value of different stimuli (i.e., reward probabilities associated with different stimuli) based on different learning strategies[33,84]. The response dissimilarity matrix was computed as the distance between the activity of recurrent excitatory or inhibitory populations during the choice period (Supplementary Fig. 9). The reward probability dissimilarity matrix was calculated as the distance between reward probability estimates based on each of the three models (informative-feature, conjunction of non-informative features, and object-based models) for all the stimuli. We then used GLMs to estimate the normalized weights of the reward probability dissimilarity matrix for predicting the response dissimilarity matrix. We obtained qualitatively similar results when measuring dissimilarity with Euclidian distance, correlation, and cosine, but only report those based on the Euclidian distance.

**Measuring reward modulations of connection weights**. To measure the effects of reward-dependent plasticity, we examined changes in the weights of plastic sensory input during the simulation of our task. Specifically, we used GLM to fit weights from a given sensory population type (e.g., yellow color-encoding unit) to a given recurrent population (e.g., $\text{Exc}_{fr}$) using the reward probability estimates associated with a feature or conjunction of features represented by that sensory population (e.g., value of color yellow). Value estimates were calculated using a simple reinforcement learning agent (learning rates ($\alpha_{rew}, \alpha_{unr}$) = 0.05). Additionally, we used GLMs to estimate changes in plastic recurrent connection weights as a function of time (trial number).

**Data analysis**. Unless otherwise mentioned, the statistical comparisons were performed using Wilcoxon signed-rank test in order to test the hypothesis of zero median for one sample or the difference between paired samples. The reported effect sizes are Cohen's $d$-values. All data was collected using custom code written in Matlab with Psychtoolbox Version 3 extensions[85]. All behavioral analyses and model fitting were done using custom code written in MATLAB 2018a (Math-Works, Inc., Natick, MA). RNN simulations were implemented using the Tensorflow package in Python 3.7 environment[86].

**Reporting summary**. Further information on research design is available in the Nature Research Reporting Summary linked to this article.

## Data availability

The raw and processed data generated in this study have been deposited in a GitHub repository accessible at https://doi.org/10.5281/zenodo.5594684[87]. Source data are provided with this paper.

## Code availability

The source code to reproduce the results of this study has been deposited in a public repository accessible at https://doi.org/10.5281/zenodo.5594684[87].

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

## Acknowledgements

This work is supported by the National Science Foundation (CAREER Award BCS1943767 to A.S.). We would like thank Jane Xu for collecting some of the experimental data and Venice Nomof and Zohra Aslami for collecting data in an earlier version of the experiment.

## Author contributions

A.S. and S.F. designed the study. S.F. performed the experiments. A.S. and S.F. analyzed the experimental data. S.F. designed the model, performed model simulations, and analyzed the data. A.S. and S.F. wrote the paper. A.S. and S.F. contributed to revising the manuscript.

## Competing interests

The authors declare no competing interests.
