## [Peer Review File · Nature Communications]

Computational mechanisms of distributed value representations and mixed learning strategiesREVIEWER COMMENTS

Reviewer #1 (Remarks to the Author):

In this paper the authors present a behavioural study on how humans make decisions when there are multiple stimuli available in the environment. The paper further develops an RNN model to investigate the mechanisms behind the behavioural processes. The paper is overall interesting and well-written. The task is also interesting and targets an important question about human decision-making. However, I believe further analyses are required to confirm the reported behaviour effects and also further comparisons with neural recordings are required to establish the usefulness of the developed RNN. Based on this, I have some major concerns about the paper as detailed below.

- Behavioural analyses. The plots and analyses shown to support behavioural claims of the paper are well prepared and interesting, but it is still unclear to me whether the effects (model neutral effects) are significant at the group level (without subdividing subjects into feature/conjunction learners). Based on this, I suggest adding three further analyses. Firstly, showing two graphs which show the probability of choosing the same feature (or conjunction) in the next trial, depending on whether the previous trial was rewarded/not rewarded (figure 2 shows this to some extent, but group averages are not clear and further the groupings are not model neutral and depend on model fits). Secondly, I suggest (in line with the previous graphs) conducting a logistic GLM analysis on the choices (currently it is on probability estimates) investigating the effect of reward and its interactions with features/conjunctions on staying on the same action (feature/conjunction) in the next trial (for all subjects). The third analysis I would suggest is to show the difference between DR_informative and DR_non information for feature and conjunction on the same plot (as x and y axes) for each subject, so that it can be shown how much each subject uses both types of learning (feature and conjunction). From the current plots the degree to which both strategies are used by the same person is unclear.

- RNN fit to data. Although figure 4 and 1 are similar, it is unclear from the current analyses how much RNN simulations produce *choices* similar to humans. This is because the suggested RNN model lacks a choice layer mechanism and only predicts reward probabilities. To address this limitation, I would suggest adding a choice layer (e.g., a linear layer that reads out the output activities of the network), simulating the RNN and performing the above analysis (or the one in Fig 2) to show much the effect of reward on actions in RNN simulations is similar to that of the subjects.

- In terms of behaviour, although the RNN models are consistent with what was found in the previous analysis, they didn't really provide much novel insights about behaviour. In this regard, I would suggest at least comparing the fit of RNN to behaviour to the baseline RL models (e.g., using cross-validation for predicting choices, similar to Ref 3 below) to show that the baseline models capture the same amount of variance that the RNN models can capture.

- The sections about analysing the structure of weights in RNN are well presented and well developed, but without showing (preferably quantitatively) how well the developed mechanisms track actual neural activities/plasticity, it would be hard to assess the importance of the findings. Is it possible for example to show the activities/plasticities are related to the brain activities in some specific conditions/task? (e.g., similar to Ref 1 below).

- I think further discussions about the literature on category learning (about feature and conjunction learning) would be appropriate (beyond generalisation aspects, which are currently discussed). Also, the current RNN training setup is similar to learning to learn (or meta-learning) literature in machine learning (e.g., Ref 2 below), which have been previously applied to human choice data (e.g., Ref 3 below). The authors can consider adding a discussion of this relevant literature.

- The differences in BIC (Supple Table 1) is marginal between F+C1 and Feature-based models for coupled and uncoupled cases, but become significantly larger in the Decay models. Is there justification for why the decay parameter should affect F+C1 and Feature-based models differently?

Minor:

- Please provide information about how the parameters of RNN were chosen (for example λ_r in equation 19).

-I would suggest bringing the description of the task to the main text.
tensor flow => Tensorflow. Also please add citations.

Ref 1:

Sussillo, David, et al. "A neural network that finds a naturalistic solution for the production of muscle activity." *Nature neuroscience* 18.7 (2015): 1025-1033.

Ref(s) 2:

Wang, Jane X., et al. "Prefrontal cortex as a meta-reinforcement learning system." *Nature neuroscience* 21.6 (2018): 860-868.

Wang, Jane X., et al. "Learning to reinforcement learn." *arXiv preprint arXiv:1611.05763* (2016).

Ref 3:

Dezfouli, Amir, et al. "Models that learn how humans learn: the case of decision-making and its disorders." *PLoS computational biology* 15.6 (2019): e1006903.

Reviewer #2 (Remarks to the Author):

Farashahi S and Soltani A, Neural mechanisms of distributed representations and learning strategies

The authors describe a study in which they carried out behavioral studies in human subjects, and trained an RNN on a similar task. The task required subjects to learn to select multidimensional cues that predicted reward. Cues varied in shape, color, and texture. They found that, although challenging, subjects could learn the task, and most learned both shape and feature conjunctions that predicted reward. The behavior of these subjects was well-fit by an RL model which also included these factors. The trained RNN was also able to learn to predict outcome value. They also examined value coding as a function of cue features for hidden units in the RNN that did or did not have plasticity and/or were inhibitory vs. excitatory. They found that plasticity of connections was important for developing value representations.

This is an interesting study which attempts to identify computational mechanisms that may underlie learning value associations for high dimensional stimuli. The results as presented are detailed, clear and straightforward. A number of choices in the modeling were not clearly motivated, however, and additional clarification of why these choices were made, as well as how important they are, is important.

Comments

1. It would be best to incorporate Supplemental Fig. 1 into the main text so the task design is clear.
2. The network was trained in two stages. Why was this? What if the network was trained directly on the value estimation task given to the subjects, as opposed to the object based version? I assume it is hard to get the network to learn this and that is why the alternative approach was used. But this should be demonstrated and clarified.
3. How does the network perform on samples on which it was not trained? What if it was trained on a subset of cues and then used to predict on the other set? Some generalization, for example across the same shape with different colors, should be fine. This is important as the network may be overfitting.
4. The R2 between the network and the RL model was higher than it was between the network and

the subjects. It would be useful to add noise to the network in some way, to better match performance between the network and the human participants.

5. I would suggest not calling this network biophysically plausible. Maybe you could say that the network has some biophysically realistic features.

6. What if the network input was simplified to just the shape features? The input dimensionality was quite high. I also think that if one was equating this input dimensionality to visual areas, it might represent inputs from multiple areas.

7. It would be useful to directly illustrate the representational similarity analysis. I would show a matrix showing the value mapping for each stimulus, and then another matrix for activity levels for an example population, and then show the regression (GLM) used to map between these for an example. The slope can then be illustrated, and it can be shown that this is what is being analyzed.

8. It would be useful to use dimensionality reduction on the neural dynamics at the beginning and end of learning, to show that the trajectories for two cues that differ in value maybe do not differ at the beginning of training, but then differ at the end.

9. How much variance in the value estimates are driven by shape vs. feature conjunction? Ultimately the authors are carrying out non-linear regression with a basis function network, so it can certainly approximate the function mapping from cue to value. One would expect that the amount of variance in the unit activity should match the amount of variance in the relationship between feature dimension and value.

10. Why was the network simulated to have plasticity in only certain connections? Is there any evidence for this in any neural systems? The choices were not well motivated by the biology.

Reviewer #3 (Remarks to the Author):

This study looks at how complex learning strategies might be implemented in humans and Recurrent Neural Networks (RNNs). Specifically, they look at a learning scenario in which the agent needs to learn about the informativeness of features and can additionally exploit conjunction and object level information to improve performance. Humans and RNN's learn this in a characteristic way, first abstracting for feature dimensions and then adding conjunction level information. RNN's do this through plasticity at the "sensory" i.e. stimulus identity level, which also contain all the reward sensitivity, with a particular relevance of plastic inhibitory neural pools and their connections to plastic excitatory ones, as shown using RNN lesions. Additionally, plastic inhibitory neurons appear to contain feature-based information and excitatory ones more object based information. All these results are very interesting and open up many new potential empirical tests in actual neural populations. However, the one thing that is lacking a little is more of an argument why the results are the way they are. E.g. did the authors expect excitatory populations to have this link to object based learning and inhibitory with feature based learning? Is it because of the fact that only excitatory populations have direct links to outputs or could there be another reason?

Overall, there is a lot to like in this manuscript. It contains extensive exploration of the properties of the RNN, has an interesting task and compelling correspondence between human and model results. If I had to criticize the manuscript, I would have liked a little bit more conceptual reasoning about why the results are how they are and a bit more clarity in some places. Otherwise, congratulations on an impressive study!

Major comments:

1) The task itself should be in main figures. Also there could additionally be a bit more clarity from the start about the paradigm.

2) Figure 6 C talks about all 8 types. Why are there only the 4 excitatory shown? I assume the 8 was a mistake because from the rest of the manuscript it is clear that only the excitatory neurons are connected to the output.

3) The authors should more clearly state that the statistical values for the model free differential response after sorting participants into conjunction+feature and feature only learners are not unbiased as they are based on model fits of the same data. I think the description of the tests are still informative but that there is a differential response to informative features is not surprising when it is a test in the participants that were fit best by Conjunction+feature, unless I am misunderstanding something about the procedure.

4) The authors convincingly show that there is in their own words "... an opponency between representations of feature and object values by excitatory and inhibitory neurons;" But what I am not quite sure of is the why. Is it because of the fact that only excitatory populations have direct links to outputs or could there be another reason? It would be nice to have more discussion of this by the authors.

5) There is a discrepancy between the RNN results regarding the excitatory plastic population which focuses on object level learning and the people doing conjunction and feature based learning. The authors need to address this a bit more head-on and explain how the narrative of combined feature and conjunction based learning through interactions between excitatory plastic and inhibitory plastic units lines up with the fact that there are object-based effects in excitatory populations and object based learning emerges after lesioning in the model despite no object based learning seeming existing in the human participants.

6) Related to the last point, from Figure 1 B it looks like later in learning feature based learning does worse than object based learning, is that correct? I am asking in part because in Fig 1 C this doesn't seem to be the case for the R-squared measure, although it seems to for the goodness of fit. Clarification on this is appreciated (and whether adding object based information explains additional variance which would also explain what the object-based excitatory effects in the RNN might be doing).

7) A lot of the analysis of the RNN are descriptive and exploratory (e.g. they authors simply test all sub populations for significant weight change and report the one that changes). This makes a lot of sense because they want to give an exhaustive description of what the model is doing, which I applaud. However, it wasn't always clear how the authors take into account multiple comparison issues in those cases? If they haven't, they could simply re-run the simulation (if that is possible) and this way confirm all results. Alternatively, in cases where they had no prior hypothesis on what population should change, they could adjust statistics for multiple comparisons and otherwise clearly state that they aren't doing such adjustments or what the prior hypothesis was and why.

Minor:

A) Does Fig 1 D mean conjunctions are used a lot less or are weights not comparable like that across types?

B) Fig 1D hard to see difference between grey and black at first.

C) In Fig 6 D there are 3 columns for feature but only one is significant. If this is the informative feature, please make it clearer in the figure itself.

D) Fig 3 C. Are inhibitory neurons always mostly active really early on and then silenced or is this just in that specific trial? Also, out of curiosity, why are there no other descriptions of timecourses and neural dynamics? Were they not informative?

E) Fig 4 A could use a legend in figure.

F) How would you have to change the RNN to get the feature only learning as some people (1/3) do?

G) Maybe have labels in Fig 5 (even though it wont look as nice, people might be able to follow even without remembering how the images work). Alternatively, add a legend to remind them?

H) In the analysis of the human data, differential response for non-informative and informative features should be compared against each other if the authors want to make a comparative statement, not each against zero (this seemed to be done with feature learners but not feature and conjunction learners).

I) The authors state "In this step, only connections endowed with the plasticity mechanism were modulated after receiving reward feedback in each trial. The overall task structure used in these simulations was similar to our experimental paradigm with a simple modification where only one stimulus was shown in each trial and the network had to learn the reward probability associated with that stimulus." This means there is no possibility for spread of effect across chosen and unchosen stimuli in the model. Was there any evidence in their participant data for such effects?

Reviewer #4 (Remarks to the Author):

The authors studied how humans and neural networks learn values in an environment where visual features and conjunctions of features are predictive of values. They showed that both humans and a type of plastic recurrent neural networks (RNNs) showed a mixed learning strategy, in which the feature-based values are learned earlier and faster, while conjunction-based values are learned later and slower. They showed these points with a combination of model fitting and direct data analysis. Then the authors proceeded with a detailed investigation of the RNNs. Through thorough analyses probing neural representation, connection weights, and lesioning impacts, they revealed how different neural types (plastic vs not, excitatory vs inhibitory) in the model are involved in learning the values.

This work is carefully done, with plenty of controls and complementary analyses. The paper is clearly written, and the figures are thoughtfully made. The experimental task is interesting, and the kind of recurrent neural network used is innovative. Overall, this is a very solid paper.

The main concerns I have are (1) the core finding from the human participants is, in my opinion, not so surprising, (2) the RNN part, though abundant in observations, does not provide much intuitive understanding.

I want to emphasize that these concerns do not, in any way, refute the authors' findings and observations. I'm also not suggesting major additional experiments, because I think the issue is not a lack of controls or care. The authors have done a lot of work, and in my opinion, deserve to have this paper published with few additional edits at a respectable journal.

Major points:

(1) Human behavior not surprising, or in other words, there was no clearly articulated alternatives to what the authors found.

The authors report that humans learn feature-based rules (red better than blue) faster than conjunction-based rules (blue triangle better than red circle). The authors emphasize that they are the first to report this phenomenon (e.g., "...our experimental results demonstrate for the first time..."). However, it's not clear what would be a plausible alternative. Could humans possibly learn

conjunction-based rules before they learn feature-based rules?

This is further complicated by the fact that informative conjunction occurs much less frequently than informative features. Therefore, it's even harder to imagine than conjunctions would be learned first.

I'm willing and hoping to be convinced by the authors that this behavior is not expected by classical theories or simply our everyday intuition. But currently I don't see that from the Introduction.

(2) RNN results lack a summarizing intuition.

The RNN itself is interestingly complex in its architecture (two cell types, optional input plasticity, optional recurrent plasticity), and the authors conducted a very thorough investigation of it. However, at the end of it, we are left with lots of observations of the RNN, but little intuition that may transfer to other settings.

For example, the authors observed that feature-encoding input units connect more strongly to inhibitory neurons that have plastic input weights. Why? As far as I can tell, little intuition is provided, and little investigation is oriented at understanding the difference between cell types, rather than reporting it. Some sections describing the RNN results (parts of p12-p19) almost read like auto generated text reporting all combinations of analyses (e.g. X analysis on Y cell report Z value).

I think this issue is demonstrated in the authors' Abstract where they say the "learning strategy relies on...distinct contributions of inhibitory and excitatory neurons". That sounds rather vague.

(3) Related to the second point, the architecture comparison is not given enough attention.

Toward the end of the Results section, the authors discussed what happens with alternative structures. I find this section with lots of promises, because it may help us understand why the plastic RNN is necessary and why the human findings are surprising. In particular, I'm surprised to know that some architectures didn't learn like humans do, which can be an important reason to think that the human findings are not trivial (essentially countering my first major concern). But being a short section at the very end, it feels like an afterthought and a missed opportunity.

If I assumed the authors had infinite time and patience, I would have suggested that they restructure the paper, and put more emphasis on this last section. Perhaps bring it up, and show it before the main RNN results. But I know that in reality, we all have many things to do, and it's not worth restructuring a perfectly fine paper just to satisfy one reviewer's preference. So I'm not suggesting the authors do anything about this. I'm only pointing this issue out in the hope that the authors may find it helpful for their future work.

Minor points

In the Abstract, it's not always clear what findings are based on RNNs and what are from humans.

Fig. 1: I would appreciate if there's a schematic for the task. Why not move Fig. S1 A, B to Fig. 1?

Fig. 2 and page 8, the effect for informative conjunction seems relatively weak ($P=0.029$). But I suppose it's not clear if the authors can do anything about it.

The reason for using RNN is not well articulated. Usually people use RNNs to have temporal dynamics, but Fig. 3c shows essentially no dynamics?

Fig. 7. Perhaps write something like "Lesioned networks" in the figure (not just the caption)?

Response to the reviewers' comments, and summary of changes made in response to the comments of the reviewers.

Title: Neural mechanisms of distributed value representations and mixed learning strategies

Authors: Farashahi and Soltani

We are greatly thankful to all the reviewers for their careful reading of our manuscript and for their useful and constructive suggestions. We have performed multiple additional analyses and made substantial changes in the revised manuscript to fully address all of the reviewers' concerns and suggestions. Below, we provide point-by-point responses to individual reviewers' comments and concerns. The corresponding changes have been clearly marked (in blue) and noted in the revised manuscript (e.g., [R1.1] indicates response 1 to Reviewer # 1' comment, etc.).

Reviewer #1

"In this paper the authors present a behavioural study on how humans make decisions when there are multiple stimuli available in the environment. The paper further develops an RNN model to investigate the mechanisms behind the behavioural processes. The paper is overall interesting and well-written. The task is also interesting and targets an important question about human decision-making. However, I believe further analyses are required to confirm the reported behaviour effects and also further comparisons with neural recordings are required to establish the usefulness of the developed RNN. Based on this, I have some major concerns about the paper as detailed below."

Response: We thank the reviewer for a positive evaluation of our work. We really hoped that there were existing neural recordings that could be compared with our modeling results and used to test our predictions directly. However, because of the novelty of our experimental paradigm, there is no such neural data. Nonetheless, we hope that our additional new analyses and answers provided here, and corresponding changes in the revised manuscript, address the rest of the reviewer's concerns.

"Major:

1. Behavioural analyses. The plots and analyses shown to support behavioural claims of the paper are well prepared and interesting, but it is still unclear to me whether the effects (model neutral

effects) are significant at the group level (without subdividing subjects into feature/conjunction learners). Based on this, I suggest adding three further analyses.

1a. Firstly, showing two graphs which show the probability of choosing the same feature (or conjunction) in the next trial, depending on whether the previous trial was rewarded/not rewarded (figure 2 shows this to some extent, but group averages are not clear and further the groupings are not model neutral and depend on model fits)."

Response: We thank the reviewer for their comment and useful suggestion. To address this, we now have calculated and compared differential response for the informative and non-informative features and the informative and non-informative conjunctions across all participants.

We found that the difference between the probability of selecting stimuli that contained the informative feature of the stimulus that was selected and rewarded in the previous trial and the same probability when the previous trial was not rewarded (i.e., differential response of the informative feature) was significantly larger than zero (two-sided sign-rank test; $P = 10^{-3}$, $d = 0.61$, $N = 67$; Figure 1A below). In contrast, the difference between the probability of selecting stimuli that contained the non-informative features of the stimulus selected and rewarded in the previous trial and the same probability when the previous trial was not rewarded (i.e., differential response of the non-informative features) was not significantly different than zero (two-sided sign-rank test; $P = 0.09$, $d = 0.28$, $N = 67$; Figure 1B below). These results indicate that participants responded differently to reward vs. no reward depending on the informativeness of features of the selected stimulus in the previous trial.

A similar analysis on the informative and non-informative conjunctions yielded consistent results. More specifically, we found that the difference between the probability of selecting stimuli that contained the informative conjunction of the stimulus selected and rewarded in the previous trial and the same probability when the previous trial was not rewarded (i.e., differential response of the informative conjunction) was significantly larger than zero (two-sided sign-rank test; $P = 2.7 \times 10^{-3}$, $d = 0.65$, $N = 67$; Figure 1C below). In contrast, the corresponding difference for the non-informative conjunctions (i.e., differential response of the non-informative conjunctions) was not significantly different than zero (two-sided sign-rank test; $P = 0.11$, $d = 0.21$, $N = 67$; Figure 1D below). These results indicate that participants responded differently to reward vs. no reward depending on the informativeness of conjunctions of the selected stimulus in the previous trial.

Figure 1. Response to reward feedback depends on the informativeness of a feature or conjunction of features of the selected stimulus. (A) Plotted is the probability of selecting stimuli that contained the informative feature of the stimulus that was selected and rewarded (R) in the previous trial versus the same probability when the previous trial was not rewarded (NR). The insets show the histogram of the difference between these two probabilities (i.e., differential response). The dashed lines show the median values across participants, and an asterisk indicates the median is significantly different from 0 using two-sided sign-rank test with $P < 0.05$. (B) Similar to (A) but for stimuli that contained the non-informative features of the stimulus that was selected and rewarded in the previous trial versus the same probability when the previous trial was not rewarded. (C–D) Similar to (A–B) but for the informative or non-informative conjunctions.

We now have added Figure 1 above as **Supplementary Figure 5** and discussed its results in the revised manuscript (see [R1.1] in the revised manuscript).

“1b. Secondly, I suggest (in line with the previous graphs) conducting a logistic GLM analysis on the choices (currently it is on probability estimates) investigating the effect of reward and its interactions with features/conjunctions on staying on the same action (feature/conjunction) in the next trial (for all subjects).”

Response: We thank the reviewer for this suggestion. There are a few issues with applying GLM to choice data, and resolving these issues would give very similar results to the above analyses. First, in each trial, each of the two presented stimuli contains two out of three values of each feature and two out of nine values of each conjunction. This means that more than one feature (or similarly more than one conjunction) can be present in consecutive trials. Therefore, depending on the feature (or conjunction) of interest, participants' choice can be interpreted differently. Second, in some trials, participants may have no choice but to choose the same feature (or same conjunction) as in the preceding trial. This means that not all trials can be used to investigate the effect of obtained reward in the previous trial based on the "free" choice of staying on the same feature (or conjunction). To overcome these issues, one can limit the analysis to trials in which only one of the two stimuli contains the same feature as the feature of chosen stimulus in the preceding trials (similarly for conjunctions). However, this results in having separate GLMs for each feature (or conjunction) applied to different subsets of trials. Such an analysis is very similar in nature to the differential response analysis mentioned above, which is now reported in the revised manuscript.

"1c. The third analysis I would suggest is to show the difference between DR_informative and DR_non information for feature and conjunction on the same plot (as x and y axes) for each subject, so that it can be shown how much each subject uses both types of learning (feature and conjunction). From the current plots the degree to which both strategies are used by the same person is unclear."

Response: We thank the reviewer for this helpful suggestion. To address this, we have calculated differential response for non-informative conjunctions and subsequently compared differential response of the informative and non-informative features, and differential response of the informative and non-informative conjunctions.

We found that differential response for the informative feature and differential response for the informative conjunction were both positive for participants who adopted the best mixed feature- and conjunction-based learning strategy, the F+C₁ model (two-sided sign-rank test; informative feature: $P = 8 \times 10^{-4}$, $d = 0.59$, $N = 41$; informative conjunction: $P = 0.029$, $d = 0.19$, $N = 41$). In contrast, differential responses for the non-informative features and the non-informative conjunctions were not distinguishable from 0 for the same participants (two-sided sign-rank test; non-informative features: $P = 0.23$, $d = 0.058$, $N = 41$; non-informative conjunctions: $P = 0.47$, $d = 0.056$, $N = 41$). We also found that

differential response for the informative feature was larger than that of the non-informative features (two-sided sign-rank test; $P = 0.013$, $d = 0.41$, $N = 41$; Figure 2A below). Similarly, differential response for the informative conjunction was larger than that of the non-informative conjunctions (two-sided sign-rank test; $P = 0.031$, $d = 0.38$, $N = 41$; Figure 2B below). Finally, the difference between differential response for the informative and non-informative features was larger than the difference between differential response for the informative and non-informative conjunctions (two-sided sign-rank test; $P = 0.025$, $d = 0.23$, $N = 41$; Figure 2C below).

Figure 2. Direct evidence for adoption of mixed feature- and conjunction-based learning strategy. (A) Plot shows differential response for the informative feature vs. differential response for the non-informative features for participants whose choice behavior was best fit by the F+C₁ model. The inset shows the histogram of the difference between differential response of the informative and non-informative features. The dashed line shows the median values across participants, and the asterisk indicates the median is significantly different from 0 using two-sided sign-rank test with $P < 0.05$. (B) Plot shows differential response for the informative conjunction vs. differential response for the non-informative conjunctions for the same participants. The inset shows the histogram of the difference between differential response of the informative and non-informative conjunctions. (C) Plot shows the difference between differential response for the informative and non-informative features vs. the difference between differential response for the informative and non-informative conjunctions. The inset shows the histogram of the difference between the aforementioned differences. (D–E) Similar to (A–C) but for participants whose choice behavior was best fit by the feature-based model.

In contrast, for participants who adopted the feature-based learning strategy, only differential response for the informative feature was significantly larger than zero (two-sided sign-rank test; informative feature: $P = 0.022$, $d = 0.37$, $N = 21$; non-informative feature: $P = 0.09$, $d = 0.25$, $N = 21$), with differential response for the informative feature being larger than that of non-informative features (two-sided sign-rank test; $P = 0.005$, $d = 0.43$, $N = 21$; Figure 2D above). Moreover, for these participants, differential response for either the informative conjunction or non-informative conjunctions was not distinguishable from zero (two-sided sign-rank test; informative conjunction: $P = 0.27$, $d = 0.17$, $N = 21$; non-informative conjunctions: $P = 0.08$, $d = 0.41$, $N = 21$) and differential response for the informative conjunction was not distinguishable from that of non-informative conjunctions (two-sided sign-rank test; $P = 0.15$, $d = 0.12$, $N = 21$; Figure 2E above). Finally, we found that the difference between differential response for the informative and non-informative features was larger than the difference between differential response for the informative and non-informative conjunctions (two-sided sign-rank test; $P = 0.045$, $d = 0.31$, $N = 41$; Figure 2F above).

Together, these results dovetail our previous findings, providing direct evidence for adoption of different learning strategies. In the revised manuscript, we now have replaced previous **Figure 2** (now **Figure 3**) with Figure 2 above and discussed its results (see [R1.2]).

*“2. RNN fit to data. Although figure 4 and 1 are similar, it is unclear from the current analyses how much RNN simulations produce *choices* similar to humans. This is because the suggested RNN model lacks a choice layer mechanism and only predicts reward probabilities. To address this limitation, I would suggest adding a choice layer (e.g., a linear layer that reads out the output activities of the network), simulating the RNN and performing the above analysis (or the one in Fig 2) to show much the effect of reward on actions in RNN simulations is similar to that of the subjects.”*

Response: We thank the reviewer for raising this concern. The reviewer is correct that the presented RNNs do not make any decisions. This simplification was adopted to better focus on learning aspects of the task. Nonetheless, to address reviewer’s concern, we added a decision layer (using a logistic function) to the output layer of RNNs to generate choice based on the presented pair of stimuli in each trial. We then compared choices made by RNNs with our experimental data as described in the next response (see our response to major point 3). To compute differential response of RNNs, however, we still

used estimated reward probabilities in the output layer to avoid unnecessary additional simulations needed to compensate for stochasticity in generation of choice in each trial.

Analyses of differential response of RNNs yielded results similar to those of our human participants. Specifically, we found that in the trained RNNs, differential responses for the informative feature and informative conjunction were both positive (two-sided sign-rank test; informative feature: $P = 0.03$, $d = 0.28$, $N = 50$; informative conjunction: $P = 0.04$, $d = 0.23$, $N = 50$). In contrast, differential response for the non-informative features and the non-informative conjunctions were not distinguishable from 0 (two-sided sign-rank test; non-informative features: $P = 0.11$, $d = 0.14$, $N = 50$; non-informative conjunctions: $P = 0.13$, $d = 0.09$, $N = 50$).

Figure 3. Trained RNNs can replicate main behavioral results in terms of response to reward feedback. (A) Plot shows differential response for the informative feature vs. differential response for the non-informative features in the trained RNNs. The inset shows the histogram of the difference between differential response of the informative and non-informative features. The dashed line shows the median values across all the trained RNNs, and the asterisk indicates the median is significantly different from 0 using two-sided sign-rank test with $P < 0.05$. (B) Similar to (A) but for the informative and non-informative conjunctions. (C) Plot shows the difference between differential response for the informative and non-informative features vs. the difference between differential response for the informative and non-informative conjunctions. The inset shows the histogram of the difference between the aforementioned differences.

In addition, differential response for the informative feature was larger than that of the non-informative features (two-sided sign-rank test; $P = 0.013$, $d = 0.41$, $N = 50$; Figure 3A above). Similarly, differential response for the informative conjunction was larger than that of the non-informative conjunctions (two-sided sign-rank test; $P = 1.4 \times 10^{-6}$, $d = 0.48$, $N = 50$, Figure 3B above). Finally, similar to our experimental results, the difference between differential response for the informative and non-informative features was larger than the

difference between differential response for the informative and non-informative conjunctions (two-sided sign-rank test; $P = 0.018$, $d = 0.18$, $N = 50$; Figure 3C above). Together, these results illustrate that the trained RNNs can qualitatively replicate behavior of human participants.

We now have discussed the above results in the revised manuscript and included Figure 3 above as a part of **Figure 5** (see [R1.3]).

“3. In terms of behaviour, although the RNN models are consistent with what was found in the previous analysis, they didn’t really provide much novel insights about behaviour. In this regard, I would suggest at least comparing the fit of RNN to behaviour to the baseline RL models (e.g., using cross-validation for predicting choices, similar to Ref 3 below) to show that the baseline models capture the same amount of variance that the RNN models can capture.”

Response: We thank the reviewer for this suggestion. Unfortunately, we cannot fit RNNs to behavior as suggested because the number of free parameters in our network exceeds the number of choice trials we obtained from our participants. Specifically, a network with similar structure to our proposed RNNs that is able to perform our experiment successfully would have a total of 29862 free parameters (including naïve input, naïve recurrent and naïve output weights, bias terms, and Hebbian learning parameters) compared to the total of 28944 trials we obtained from our participants.

Nonetheless, to show the behavioral relevance of our RNNs, as suggested by the reviewer, we added a decision layer (using a logistic function) to the output layer of RNNs to generate choice based on the presentation of a pair of stimuli in each trial. We then fit choice data produced by the trained RNNs using different RL models. Specifically, we used coupled, uncoupled, and decay RL models that were shown to capture our participants’ choice behavior well (Table 1 below). We found that similar to our human participants, the mixed feature- and conjunction-based model provided the best fit to choice data from the trained RNNs.

In the revised manuscript, we now have discussed the above results and added Table 1 below as **Supplementary Table 2** (see [R1.4]).

Model	Feature-based	Mixed feature- and conjunction-based			Object-based	Mixed feature- and object-based			
		F+C ₁	F+C ₂	F+C ₃		F ₁ +O	F ₂ +O	F ₃ +O	
# pars.	5	6	6	6	3	6	6	6	
-LL	167.5±3.4	163.4±3.4	168.8±3.3	169.1±3.3	173.0±3.0	169.7±3.5	171.7±3.6	172.5±3.7	Coupled
AIC	345.1±6.8	337.9±6.9	349.5±6.7	350.9±6.6	352.0±6.2	346.3±6.9	343.4±7.2	351.0±7.5	
BIC	363.1±6.8	360.6±6.9	371.1±6.7	372.4±6.6	362.8±6.2	368.9±6.9	371.0±7.2	372.6±7.5	
-LL	167.1±3.4	162.9±3.4	167.9±3.2	168.3±3.0	172.6±3.0	169±3.7	170.4±3.6	172.0±3.6	Uncoupled
AIC	344.3±6.7	336.9±6.8	347.7±6.5	348.7±6.2	351.2±6.0	345.9±7.4	346.8±7.2	350.1±7.1	
BIC	362.3±6.7	359.6±6.8	369.3±6.5	370.3±6.2	362.0±6.0	367.5±7.4	368.4±7.2	371.7±7.1	
# pars.	6	7	7	7	4	7	7	7	
-LL	167.2±3.5	161.7±3.4 **(**)	167.3±3.0	170.0±3.0	171.8±2.8	166.8±3.6	167.4±3.6	167.9±3.6	Decay
AIC	346.4±6.9	337.4±6.7 *(**)	348.7±6.2	354.0±6.0	351.7±5.6	347.7±7.2	348.9±7.2	349.9±7.3	
BIC	368.0±6.9	362.6±6.7 *(**)	373.9±6.2	379.2±6.0	366.0±5.6	372.9±7.2	374.0±7.2	375.0±7.3	

Table 1. Comparison of the goodness-of-fit measures for fitting choice data generated by the trained RNNs. Reported are the goodness-of-fit measures, negative log likelihood (-LL), Akaike information criterion (AIC), and Bayesian information criterion (BIC) averaged over all the trained RNNs (mean±s.e.m.). The model providing the best fit (F+C₁) and its object-based and feature-based counterparts are highlighted in green and orange, respectively. The symbols next to goodness-of-fit for the F+C₁ model indicate comparison with the feature-based and object-based (shown in parenthesis) models using a two-sided, sign-rank test. The significance level of the test is coded as: $0.01 < P < 0.05$ (*), $0.001 < P < 0.01$ (**), and $P < 0.001$ (***) .

“4. The sections about analyzing the structure of weights in RNN are well presented and well developed, but without showing (preferably quantitatively) how well the developed mechanisms track actual neural activities/plasticity, it would be hard to assess the importance of the findings. Is it possible for example to show the activities/plasticities are related to the brain activities in some specific conditions/task? (e.g., similar to Ref 1 below).”

Response: We thank the reviewer for this comment. We really hoped that there were existing neural datasets that we could use to test the predictions of our model. However, our experimental paradigm that allows us to examine the emergence of complex learning

strategies is novel and has not been used in any experiments. In addition, we are not aware of neural recording or brain imaging in a similar task. The closest task to our experimental paradigm is that of Oemisch et al. (2019), which was inspired by our previous study (Farashahi et al, 2017). Even data from that experiment still cannot be used to address different learning strategies and instead, can only provide some indirect support for our predictions as mentioned in the Discussion.

We believe that one of the important contributions of our study is to provide very clear predictions that can be tested in future experiments.

Oemisch, M., Westendorff, S., Azimi, M., Hassani, S. A., Ardid, S., Tiesinga, P., & Womelsdorf, T. (2019). Feature-specific prediction errors and surprise across macaque fronto-striatal circuits. *Nature Communications*, 10(1), 176.

“5. I think further discussions about the literature on category learning (about feature and conjunction learning) would be appropriate (beyond generalization aspects, which are currently discussed). Also, the current RNN training setup is similar to learning to learn (or meta-learning) literature in machine learning (e.g., Ref 2 below), which have been previously applied to human choice data (e.g., Ref 3 below). The authors can consider adding a discussion of this relevant literature.”

Response: We thank the reviewer for pointing out this relevant literature.

In the revised manuscript (see [R1.5]), we have added further discussion to link our results to category learning and meta-learning:

(page 26) *“In general, feature-based and conjunction-based strategies can be considered as rule-based category learning, whereas an object-based strategy can be considered as procedural-learning [R1.5].”*

“Our training algorithm was designed to allow the network to learn a general solution for learning reward probabilities in multi-dimensional environments. Networks capable of generalizing to new tasks or environments have been the focus of the meta-learning field (Finn et al., 2017; Hospedales et al., 2020; Pfahringer et al., 2000; Thrun & Pratt, 2012; J. X. Wang et al., 2016) and were used to simulate learning a distribution of tasks (J. X. Wang et al., 2018). Extending this approach to a novel task, our modeling results thus suggest that the brain’s ability to generalize might arise from principled learning rules along with structured connectivity patterns [R1.5].”

“6. The differences in BIC (Supple Table 1) is marginal between F+C1 and Feature-based models for coupled and uncoupled cases, but become significantly larger in the Decay models. Is there justification for why the decay parameter should affect F+C1 and Feature-based models differently?”

Response: We thank the reviewer for asking this question. The decay of estimated reward probabilities aims to capture forgetting of all values that were not updated (i.e., increased or decreased due to reward feedback) in a given trial. As a result, in different models, this decay happens for different numbers of unchosen features, conjunctions, or stimuli. More specifically, after each trial, reward probabilities can decay for 6 out of 9 individual features, 24 out of 27 possible conjunctions, and 26 out of 27 individual stimuli. Therefore, depending on the base model, the decay mechanism has different effects on estimated reward probabilities and thus, the goodness-of-fit measures.

“Minor: 7. Please provide information about how the parameters of RNN were chosen (for example λ_r in equation 19).”

Response: We thank the reviewer for pointing out these missing pieces of information, which are now added to the Methods (see [R1.6]).

“8. I would suggest bringing the description of the task to the main text.”

Response: We thank the reviewer for this suggestion. As suggested by the reviewer, we now have included the previous Supplementary Figure 1 as **Figure 1** in the main text (see [R1.7]).

“9. tensor flow => Tensorflow. Also please add citations.”

Response: We thank the reviewer for pointing out this typo and the missing reference, both of which have been fixed in the revised manuscript (see [R1.8]).

“Ref 1: Sussillo, David, et al. “A neural network that finds a naturalistic solution for the production of muscle activity.” Nature neuroscience 18.7 (2015): 1025-1033.

Ref(s) 2: Wang, Jane X., et al. “Prefrontal cortex as a meta-reinforcement learning system.” Nature

neuroscience 21.6 (2018): 860-868.

Wang, Jane X., et al. "Learning to reinforcement learn." *arXiv preprint arXiv:1611.05763* (2016).

Ref 3: Dezfouli, Amir, et al. "Models that learn how humans learn: the case of decision-making and its disorders." *PLoS computational biology* 15.6 (2019): e1006903."

Reviewer #2:

"The authors describe a study in which they carried out behavioral studies in human subjects, and trained an RNN on a similar task. The task required subjects to learn to select multidimensional cues that predicted reward. Cues varied in shape, color, and texture. They found that, although challenging, subjects could learn the task, and most learned both shape and feature conjunctions that predicted reward. The behavior of these subjects was well-fit by an RL model which also included these factors. The trained RNN was also able to learn to predict outcome value. They also examined value coding as a function of cue features for hidden units in the RNN that did or did not have plasticity and/or were inhibitory vs. excitatory. They found that plasticity of connections was important for developing value representations.

This is an interesting study which attempts to identify computational mechanisms that may underlie learning value associations for high dimensional stimuli. The results as presented are detailed, clear and straightforward. A number of choices in the modeling were not clearly motivated, however, and additional clarification of why these choices were made, as well as how important they are, is important. "

Response: We thank the reviewer for a positive evaluation of our work. We hope that our new analyses, answers provided here, and corresponding changes in the revised manuscript addressed all the reviewer's concerns.

"1. It would be best to incorporate Supplemental Fig. 1 into the main text so the task design is clear."

Response: We thank the reviewer for this helpful suggestion. We now have included previous Supplementary Figure 1 as **Figure 1** in the main text (see [R2.1]).

"2. The network was trained in two stages. Why was this? What if the network was trained directly on the value estimation task given to the subjects, as opposed to the object based version? I assume

it is hard to get the network to learn this and that is why the alternative approach was used. But this should be demonstrated and clarified.”

Response: We thank the reviewer for asking this question. First, we should note that the networks were trained during the first step only (i.e., stochastic gradient descent was stopped after this step) and then were only used to perform our experiment during the second step based on reward feedback in each trial. Training in the first step, which was done in a series of multi-dimensional environments, was to allow the networks to learn a general task of learning reward probabilities. Without such training, these networks would not have the connectivity pattern necessary to learn from multi-dimensional stimuli.

We have made changes to the revised manuscript to clarify these points (see [R2.2]).

“3. How does the network perform on samples on which it was not trained? What if it was trained on a subset of cues and then used to predict on the other set? Some generalization, for example across the same shape with different colors, should be fine. This is important as the network may be overfitting.”

Response: We thank the reviewer for asking this insightful question. Indeed, training in many multi-dimensional environments with different levels of generalizability was to ensure that networks are not overfitting. Moreover, analyses of networks’ behavior showed that trained networks adopt a mixture of feature- and conjunction-based learning, which itself is generalizable (as opposed to object-based learning that is not generalizable).

Nonetheless, to directly address the reviewer’s concern about overfitting and to further explore generalization in the trained RNNs, we used the trained RNNs to perform our task but with only a subset of the stimuli (a random set of 18 stimuli out of the 27 stimuli). We then used the networks to predict the value for the subset of the stimuli not shown during learning the task (remaining 9 “leave-out” stimuli). In this way, the networks could use both feature values and conjunction values to generalize across stimuli. To test generalization, we then plotted the prediction of the network for the leave-out stimuli against their actual reward probability (see Figure 4 below). We found that estimated reward probabilities for leave-out stimuli were significantly correlated with their actual reward probabilities (spearman correlation; $\rho = 0.77$, $P = 1.7 \times 10^{-7}$), suggesting that the trained networks can generalize to stimuli that have not seen before. Moreover, our results showed that as the result of generalization, estimated reward probabilities of

leave-out stimuli can deviate from the actual reward probability of these stimuli, confirming that these networks are not overfitting.

We have added a note about generalization and overfitting to the revised manuscript and included Figure 4 below as **Supplementary Figure 6** (see [R2.3]).

Figure 4. Trained networks can generalize to stimuli that were not used during learning. Each point shows the predicted reward probability for a stimulus not used during learning vs. its actual reward probability. The dashed line shows the identity line.

“4. The R^2 between the network and the RL model was higher than it was between the network and the subjects. It would be useful to add noise to the network in some way, to better match performance between the network and the human participants.”

Response: We thank the reviewer for this suggestion. There are multiple sources of noise/variability that could increase R^2 for human participants and could result in a better match between R^2 values of RNNs and human participants. For example, spiking networks are inherently more sensitive to input noise and thus a spiking version of our trained networks could show a more similar performance to our human participants. Another source of variability could be exposure to environments with different levels of generalizability that could result in adoption of different learning strategies and thus, overall smaller R^2 values.

To show that such variability could explain some of the observed differences in R^2 , we focused on training environments with very different levels of generalizability. This was possible because reward probabilities during the training step were drawn from a uniformly random distribution between 0 and 1. To that end, we defined extreme non-generalizable environments as those that fell in the 0–20% quantile of the joint distribution

of features and individual features plus conjunctions generalizability indices (see Figure 5A below), and extreme generalizable environments as those that fell in the 80–100% quantile of the same joint distribution. We then examined the behavior of RNNs trained in these two types of environments. We found that the estimated reward probabilities by RNNs trained in generalizable environments were best fit by the feature-based model (see Figure 5B below), whereas estimates by RNNs trained in non-generalizable environments were best fit by the object-based model (see Figure 5C below).

Figure 5. Generalizability of the environments used during the training step influences the behavior of RNNs during the learning task. **(A)** The plots show the joint distribution of generalizability indices calculated for the estimated reward probabilities associated with different stimuli based on their features and combinations of individual features and conjunctions. Environments with extreme generalizability and non-generalizability used to train two different sets of RNNs are indicated with white rectangles. **(B–C)** The plots show the time course of explained variance (R^2) for reward probabilities estimated by RNNs using different learning models as indicated in the legend. The results for RNNs trained using extreme generalizable and non-generalizable environments are shown in B and C, respectively. Error bars represent s.e.m. The solid line is the average of exponential fits to RNNs’ data and the shaded areas indicate s.e.m. of the fit. Different training environments result in adopting different learning strategies and different R^2 values in our task.

More importantly, the explained variance by different models was drastically different between the two sets of RNNs (compare Figure 5B and 5C above). These results indicate that different training environments result in different R^2 values, and thus variability in training (i.e., what reward environments a network or participant has experienced before performing our task) could account for mismatch between R^2 values of human participants and RNNs.

We believe that investigating the aforementioned mechanisms to account for the observed mismatch in R^2 values is beyond the scope of this study, and thus we did not include the above results in the revised manuscript.

“5. I would suggest not calling this network biophysically plausible. Maybe you could say that the network has some biophysically realistic features.”

Response: We thank the reviewer for this suggestion. In the revised manuscript, we point out that our networks have some realistic features and use “biologically inspired” instead of “biophysically plausible” (see [R2.4]).

“6. What if the network input was simplified to just the shape features? The input dimensionality was quite high. I also think that if one was equating this input dimensionality to visual areas, it might represent inputs from multiple areas.”

Response: If we understood this point correctly, the reviewer is asking what happens if sensory input to the recurrent populations only includes populations selective to individual features and not conjunctions or object identity, the latter two corresponding to input from multiple areas. First, we do not make any assumption about whether sensory input comes from the same or different brain areas, as sensory neurons with simple (e.g., selectivity to color) or complex response selectivity (e.g., selectivity to color and orientation) can be found in the same visual areas. The only assumption here is that connections between sensory populations encoding features, conjunctions of features, or stimuli (distributed across cortical visual areas) and recurrent populations (presumably in the PFC) can be plastic to allow learning and estimation of reward values in recurrent populations. In addition, restraining input to only features seems arbitrary in our view because of what is known about connectivity and neural representations in the brain.

“7. It would be useful to directly illustrate the representational similarity analysis. I would show a matrix showing the value mapping for each stimulus, and then another matrix for activity levels for an example population, and then show the regression (GLM) used to map between these for an example. The slope can then be illustrated, and it can be shown that this is what is being analyzed.”

Response: We thank the reviewer for their constructive suggestion. As suggested by the reviewer, we prepared a new figure (Figure 6 below) to illustrate representational

similarity analysis. This figure has been added as **Supplementary Figure 8** to the revised manuscript (see [R2.5]).

Figure 6. Schematic of the representational similarity analysis (RSA). (A) The response dissimilarity matrix (DM) was computed as the Euclidean distance between the activity of recurrent populations in a certain population (e.g., Exc_{rr}) during the choice period. (B–D) The reward probability DMs were calculated as the Euclidean distance between reward probability estimates based on an object-based model (B), a model based on the conjunctions of non-informative features (i.e., the informative conjunction) (C), and a model based on the informative feature (D) for all the stimuli used in the experiment. (E) As the final step of RSA, a GLM is used to fit the response DM as a function of the normalized weights of the three reward probability DMs.

“8. It would be useful to use dimensionality reduction on the neural dynamics at the beginning and end of learning, to show that the trajectories for two cues that differ in value maybe do not differ at the beginning of training, but then differ at the end.”

Response: We thank the reviewer for this helpful suggestion on exploring the temporal dynamics of population response in our networks. To address this, we applied principal component analyses (PCA) to the response of excitatory recurrent populations of the trained RNNs because this response determines the output of the networks. More specifically, we performed three separate PCAs on the activity of excitatory recurrent populations during the simulated experiment (repeated 100 times to obtain smoother results). This includes: PCA on the response of excitatory recurrent populations to all stimuli at the beginning of each session before the network has learned about the reward environment (see Figure 7A below); PCA on the response of excitatory recurrent populations to all stimuli at the end of each session when the network has fully learned the

task (see Figure 7B below); and PCA on the response of excitatory recurrent populations to all stimuli during the choice period throughout each session (see Figure 7C below). As expected, we found that the trajectory of population response projected on three principal components was not distinguishable at the beginning of the session, whereas this response diverged according to reward value as the network learned reward probabilities.

In the revised manuscript, we now have added a few sentences to discuss these results and included Figure 7 below as **Supplementary Figure 7** (see [R2.6]).

Figure 7. Dynamics of population activity in RNNs during learning task performance. (A–B) Trajectories in the activity space formed by the first three principal components of PCA performed on the response of excitatory recurrent populations at the beginning (A) and end of each session (B). Diamonds mark stimulus onset, squares mark beginning of the choice period, and triangles mark the end of stimulus presentation. Different colors represent reward value (probability) assigned to each stimulus. (C) Trajectories in the activity space formed by the first three principal components of PCA performed on the response of excitatory recurrent populations during the choice period as the network learns about different stimuli. Larger markers indicate later trials within the session.

“9. How much variance in the value estimates are driven by shape vs. feature conjunction? Ultimately the authors are carrying out non-linear regression with a basis function network, so it can certainly approximate the function mapping from cue to value. One would expect that the amount of variance in the unit activity should match the amount of variance in the relationship between feature dimension and value.”

Response: We thank the reviewer for asking this question. Noting that stimulus identity fully determines reward probabilities (value estimates), we assume that the reviewer’s question is about variance in value estimates explained by individual features vs. conjunctions of features (i.e., they refer to individual features as “shape”). To answer this question, we first performed stepwise GLM on the actual reward probabilities to calculate

R^2 associated with reward probabilities based on the informative feature ($R^2 = 0.27$) and added R^2 due to reward probabilities based on the conjunctions of the two non-informative features ($R^2 = 0.12$). We repeated the same analysis on reward probabilities estimated by human participants and the response of the output layer of the RNNs (which determines value estimates). We then compared the results between participants and RNNs to test whether variance captured by participants and networks are similar or different (considering they experienced the same reward environment).

We found that the ratio of R^2 of the reward probability of the informative feature and the conjunctions of the other two non-informative features was not significantly different between human participants and RNNs (human participants: median±IQR = 2.52±1.19, RNNs: median±IQR = 2.68±1.02; two-sided rank-sum test; $P = 0.28$, $d = 0.12$, $N = 115$), and both ratios were not different from the ratio according to the task design (median±IQR = 0.27/0.12=2.25; two-sided rank-sum test; $P > 0.16$, $d < 0.18$, $N = 50$ for RNNs and $N = 67$ for human participants). These results indicate that the trained RNNs can capture variance in value estimates embedded in the task design similarly to human participants.

“10. Why was the network simulated to have plasticity in only certain connections? Is there any evidence for this in any neural systems? The choices were not well motivated by the biology.”

Response: We thank the reviewer for asking this question. Our aim was to be agnostic and to not make any assumptions regarding the type of connections that are modulated by reward feedback (i.e., plastic connections). This was done because sensory and recurrent populations in our RNNs could be distributed across the brain. To that end, we simulated RNNs with both plastic and non-plastic connections from input sensory populations to the recurrent populations and the connections between the recurrent populations. More specifically, for each population (excitatory or inhibitory), input connections and recurrent connections were uniformly assigned to be flexible (modulated by reward) or rigid (not modulated by reward). This uniform assignment results in the eight disjoint populations all of which were included in our networks structure.

We have clarified these points in the revised manuscript (see [R2.7]).

Reviewer #3:

“This study looks at how complex learning strategies might be implemented in humans and

Recurrent Neural Networks (RNNs). Specifically, they look at a learning scenario in which the agent needs to learn about the informativeness of features and can additionally exploit conjunction and object level information to improve performance. Humans and RNN's learn this in a characteristic way, first abstracting for feature dimensions and then adding conjunction level information. RNN's do this through plasticity at the "sensory" i.e. stimulus identity level, which also contain all the reward sensitivity, with a particular relevance of plastic inhibitory neural pools and their connections to plastic excitatory ones, as shown using RNN lesions. Additionally, plastic inhibitory neurons appear to contain feature-based information and excitatory ones more object-based information. All these results are very interesting and open up many new potential empirical tests in actual neural populations. However, the one thing that is lacking a little is more of an argument why the results are the way they are. E.g. did the authors expect excitatory populations to have this link to object based learning and inhibitory with feature based learning? Is it because of the fact that only excitatory populations have direct links to outputs or could there be another reason?

Overall, there is a lot to like in this manuscript. It contains extensive exploration of the properties of the RNN, has an interesting task and compelling correspondence between human and model results. If I had to criticize the manuscript, I would have liked a little bit more conceptual reasoning about why the results are how they are and a bit more clarity in some places. Otherwise, congratulations on an impressive study!"

Response: We thank the reviewer for their positive evaluation of our work. We hope that our additional analyses, answers provided here, and corresponding changes in the revised manuscript addressed all the reviewer's concerns.

"Major comments:

1) The task itself should be in main figures. Also, there could additionally be a bit more clarity from the start about the paradigm."

Response: We thank the reviewer for this helpful suggestion. We now have moved previous **Supplementary Figure 1** to the main text as **Figure 1** and have provided a better explanation of the paradigm (see [R3.1] in the revised manuscript).

"2) Figure 6 C talks about all 8 types. Why are there only the 4 excitatory shown? I assume the 8 was a mistake because from the rest of the manuscript it is clear that only the excitatory neurons are connected to the output."

Response: We thank the reviewer for pointing to this typo, which has been fixed in the revised manuscript (see [R3.2]).

“3) The authors should more clearly state that the statistical values for the model free differential response after sorting participants into conjunction+feature and feature only learners are not unbiased as they are based on model fits of the same data. I think the description of the tests are still informative but that there is a differential response to informative features is not surprising when it is a test in the participants that were fit best by Conjunction+feature, unless I am misunderstanding something about the procedure.”

Response: The reviewer is correct that the pattern of differential response is expected considering that subjects are divided according to the results of fitting their choice behavior. We now have clearly mentioned this point in the revised manuscript (see [R3.3]). We note that differential response analysis is an extra step we take to confirm our findings from fitting choice behavior using raw choice sequences. Nonetheless, to show direct evidence for adoption of mixed learning strategies, we performed the differential response analysis but using data from all participants. More specifically, we plotted the probability of selecting stimuli that contained the features or the conjunctions of the stimulus selected and rewarded in the previous trial versus the same probability when the previous trial was not rewarded, separately for the informative and non-informative features as well as for the informative and non-informative conjunctions across all participants (see Figure 8 below).

We found that the difference between the probability of selecting stimuli that contained the informative feature of the stimulus that was selected and rewarded in the previous trial and the same probability when the previous trial was not rewarded (i.e., differential response of the informative feature) was significantly larger than zero (two-sided sign-rank test; $P = 10^{-3}$, $d = 0.61$, $N = 67$; Figure 8A below). In contrast, the difference between the probability of selecting stimuli that contained the non-informative features of the stimulus selected and rewarded in the previous trial and the same probability when the previous trial was not rewarded (i.e., differential response of the non-informative features) was not significantly different than zero (two-sided sign-rank test; $P = 0.09$, $d = 0.28$, $N = 67$; Figure 8B below). These results indicate that participants responded differently to reward vs. no reward depending on the informativeness of features of the selected stimulus in the previous trial.

Figure 8. Response to reward feedback depends on the informativeness of a features or conjunction of features of the selected stimulus. (A) Plotted is the probability of selecting stimuli that contained the informative feature of the stimulus that was selected and rewarded (R) in the previous trial versus the same probability when the previous trial was not rewarded (NR). The insets show the histogram of the difference between these two probabilities (i.e., differential response). The dashed lines show the median values across participants, and an asterisk indicates the median is significantly different from 0 using two-sided sign-rank test with $P < 0.05$. (B) Similar to (A) but for stimuli that contained only the non-informative features of the stimulus that was selected and rewarded in the previous trial versus the same probability when the previous trial was not rewarded. (C–D) Similar to (A–B) but for the informative or non-informative conjunctions.

A similar analysis on the informative and non-informative conjunctions yielded consistent results. More specifically, we found that the difference between the probability of selecting stimuli that contained the informative conjunction of the stimulus selected and rewarded in the previous trial and the same probability when the previous trial was not rewarded (i.e., differential response of the informative conjunction) was significantly larger than zero (two-sided sign-rank test; $P = 2.7 \times 10^{-3}$, $d = 0.65$, $N = 67$; Figure 8C above). In contrast, the corresponding difference for the non-informative conjunctions (i.e., differential response of the non-informative conjunctions) was not significantly different than zero (two-sided sign-rank test; $P = 0.11$, $d = 0.21$, $N = 67$; Figure 8D above). These results indicate that participants responded differently to reward vs. no reward depending on the informativeness of conjunctions of the selected stimulus in the previous trial.

We now have added Figure 8 above as **Supplementary Figure 5** and discussed its results in the revised manuscript (see [R3.3] in the revised manuscript).

“4) The authors convincingly show that there is in their own words “... , an opponency between representations of feature and object values by excitatory and inhibitory neurons;” But what I am not quite sure of is the why. Is it because of the fact that only excitatory populations have direct links to outputs, or could there be another reason? It would be nice to have more discussion of this by the authors.”

Response: We thank the reviewer for asking this important question. To account for the observed transition between different types of learning strategies, we expected to see an opponency between representations of feature and object values. However, explaining how this opponency occurs is not straightforward as pointed out by the reviewer. Indeed, determining why and how the opponency between representations of object and feature values is materialized in the excitatory and inhibitory populations, respectively, was one of the main goals of the different simulations and analyses we performed.

As conjectured by the reviewer, the ability of excitatory recurrent populations to influence the output population could explain the larger contribution of excitatory populations to the object-based strategy (note that we did not consider any connections from inhibitory populations to the output population because inhibitory neurons are known to have predominantly local connections). More specifically, object values can be estimated directly by a single connection from sensory populations to recurrent populations because they do not require integration of information across features and/or conjunctions as is the case for feature-based and conjunction-based strategies. As a result, object values could directly drive excitatory populations which in turn drive the output of the network.

However, another important factor in driving this opponency is the interaction between excitatory and inhibitory neurons. This interaction allows for value-based modulation of the excitatory populations by the inhibitory populations whose value representations are more sensitive to different learning strategies (e.g., compare **Figure 6G** and **Figure 6C**). Specifically, our analysis of naïve input weights suggests that feature-based strategy is mainly encoded in the input connections from feature-encoding sensory populations to inhibitory populations. Additionally, because of recurrent connections, inhibitory populations disinhibit excitatory populations according to a feature-based strategy, and because our learning rules depend on pre- and post-synaptic activity,

learning in input connections from sensory populations to recurrent populations becomes dominated by the feature-based and not object-based strategy. Therefore, because of this value-dependent disinhibition, representations of feature values are reinforced (while suppressing object-based values) in excitatory populations, allowing for the intermediate strategies to emerge.

Consistently, we found that the rate of value-dependent changes in the input connections from object-encoding sensory populations to recurrent populations is not significant in the intact network (**Figure 7D**). However, once the connections from the inhibitory populations are lesioned, feature-based disinhibition is removed and the object-based strategy dominates (**Figure 8**). Therefore, the results of lesioning connections from inhibitory to excitatory populations support the idea that interactions between recurrent populations are needed to suppress object-based strategy and for the emergence of a mixed learning strategy.

We have now added some of these points to our revised manuscript (see [R3.4]).

“5) There is a discrepancy between the RNN results regarding the excitatory plastic population which focuses on object level learning and the people doing conjunction and feature based learning. The authors need to address this a bit more head-on and explain how the narrative of combined feature and conjunction-based learning through interactions between excitatory plastic and inhibitory plastic units lines up with the fact that there are object-based effects in excitatory populations and object-based learning emerges after lesioning in the model despite no object-based learning seeming existing in the human participants.”

Response: We thank the reviewer for asking this important question and mentioning an apparent discrepancy between our experimental and modeling results.

As the reviewer correctly points out, our analysis of naïve input weights and RSA suggests that object-based strategy is instantiated in and relies on excitatory neurons more strongly. If we understood the reviewer’s comment correctly, the question is why such influence of object-based strategy does not drive RNNs and result in the adoption of the object-based strategy, which is not what is observed in our experiment (please also see our response to the next point that better clarifies the influence of object-based strategy).

As explained in our response to the previous point ([R3.4]), the key to the emergence of mixed learning strategies (and suppression of an object-based strategy) is the interaction between excitatory and inhibitory populations. That is, due to recurrent connections, inhibitory populations disinhibit excitatory populations according to the

feature-based strategy. As a result of this value-dependent disinhibition, representations of feature values are reinforced (while suppressing object-based values) in excitatory populations, allowing for the intermediate strategies to drive the output layer. This is consistent with our lesioning results showing that once the connections from inhibitory to excitatory populations are lesioned and feature-based disinhibition is removed, the object-based strategy dominates the behavior.

We have now added some of these points to our revised manuscript (see [R3.5]).

“6) Related to the last point, from Figure 1 B it looks like later in learning feature-based learning does worse than object-based learning, is that correct? I am asking in part because in Fig 1 C this doesn't seem to be the case for the R-squared measure, although it seems to for the goodness of fit. Clarification on this is appreciated (and whether adding object-based information explains additional variance which would also explain what the object-based excitatory effects in the RNN might be doing).”

Response: We thank the reviewer for pointing to these seemingly contradictory results. The reviewer is correct that toward the end of the session, the fit of choice behavior by the object-based model becomes marginally better than that of the feature-based model (**Figure 2B**), but this is not the case for the explained variance in estimated reward probabilities (**Figure 2C**). We believe that this discrepancy is mainly due to similarity of object-based values to predicted values based on the F+C₁ model (spearman correlation; $\rho = 0.86$, $P = 8.3 \times 10^{-9}$), making the object-based model to fit choice behavior better than the feature-based model as participants progress through the experiment. More specifically, based on multiple analyses we have performed, it is clear that after the initial feature-based learning, participants adopt a mixed strategy using values of the F+C₁ model. Due to the similarity of estimated values in the F+C₁ model and object-based models, the goodness-of-fit for the object-based model becomes better than that of the feature-based model toward the end of the experiment.

We also ran additional analyses to provide direct evidence for this claim and show that the object-based strategy does not capture more variance than the feature-based strategy. To that end, we used stepwise GLM to fit estimated reward probabilities associated with each stimulus based on the actual reward probabilities (object-based) and the predicted reward probabilities using the informative feature and conjunction. First, we found that for both human participants and RNNs, the predicted values based on the informative feature explained most variance followed by the predicted values based on the

informative conjunction. Moreover, adding object-based values did not significantly increase the explained variance of estimated reward probabilities beyond a model that included both feature- and conjunction-based values (human participants: median±IQR = 0.9±1.0%; two-sided sign-rank test, $P = 0.12$, $d = 0.48$, $N = 67$; RNNs: median±IQR = 5.7±3.0%, two-sided sign-rank test, $P = 0.08$, $d = 0.96$, $N = 50$; Figure 9 below). Unlike fitting of choice behavior, which is done using different models separately, stepwise GLM does not suffer from the similarity of object values to predicted values based on the F+C₁ model. Together, these results demonstrate that the influence of object-based strategy on estimated reward probabilities did not increase over time and that the observed improved fit of object-based relative to feature-based models was a byproduct of the similarity of object values to predictions of the F+C₁ model (i.e., the best model).

Figure 9. Object-based values do not explain variance in estimated reward probabilities beyond the F+C₁ model. (A) Plotted is the time course of explained variance (R^2) in participants' estimates based on two GLMs: predicted values based on the informative feature and informative conjunction (F+C₁), and object values and predicted values based on the informative feature and informative conjunction (F+C₁+object). The solid line is the average of fitted exponential function to each participant's data, and shaded areas indicate s.e.m. of the fit. (B) Time course of adopted learning strategies measured by fitting participants' estimates of reward probabilities. Plotted is the weight of object values and the predicted values based on the informative feature and the informative conjunction on estimated reward probabilities. Error bars indicate s.e.m. The solid line is the average of fitted exponential function to each participant's data, and shaded areas indicate s.e.m. of the fit. (C–D) Same as (A–B) but for the trained RNNs.

We have revised the manuscript to include the aforementioned stepwise GLM and discussed its results, and we have updated other similar figures in the revised manuscript to include weights associated with object-based values (see [R3.6]).

“7) A lot of the analysis of the RNN are descriptive and exploratory (e.g. they authors simply test all sub populations for significant weight change and report the one that changes). This makes a lot of sense because they want to give an exhaustive description of what the model is doing, which I applaud. However, it wasn't always clear how the authors take into account multiple comparison issues in those cases? If they haven't, they could simply re-run the simulation (if that is possible) and this way confirm all results. Alternatively, in cases where they had no prior hypothesis on what population should change, they could adjust statistics for multiple comparisons and otherwise clearly state that they aren't doing such adjustments or what the prior hypothesis was and why.”

Response: We thank the reviewer for pointing out the issue of multiple comparisons. The reviewer is correct that correction for multiple comparisons is needed when we did not have prior hypotheses. For all analyses performed on naïve weights of the RNNs (before simulating the task), we only performed limited comparisons based on the preliminary results we obtained from $N = 20$ trained RNNs (this data was not included in data presented in the manuscript). Thus, the prior hypotheses based on our preliminary simulation data greatly reduced the number of tests we performed on the data reported in the manuscript. In the revised manuscript, we now have included correction for multiple comparisons related to naïve weights and updated all statistics accordingly. Moreover, for analyses performed on response of different types of recurrent populations and the average rate of changes in the weights of the trained RNNs, we re-ran the simulations one more time and reported results based on the new simulations. We also have mentioned our approach for correction for multiple comparisons in the Methods section of the revised manuscript (see [R3.7]).

“Minor:

8) Does Fig 1 D mean conjunctions are used a lot less or are weights not comparable like that across types?”

Response: We thank the reviewer for asking this question. The previously reported weights were actual and not normalized weights. Therefore, their values should not be compared with each other and this is why we only commented on their time course.

However, when we include all predictors based on the feature-based, feature+conjunction, and object-based models into a stepwise GLM to calculate the normalized weights of different models (see Figure 10 below), we found that indeed the normalized weight of the informative feature was significantly larger than the weight of the informative conjunction. This was reflected in the difference between the normalized weights of informative feature and conjunction being significantly larger than 0 (median±IQR = 0.28±0.08; two-sided sign-rank test, $P = 1.08 \times 10^{-5}$, $d = 0.67$, $N = 67$).

We now have clarified this point in the revised manuscript (see [R3.8]).

Figure 10. Plotted is the normalized weight of the informative feature, informative conjunction, and object (stimulus identity) on reward probability estimates. Error bars indicate s.e.m. The solid line is the average of fitted exponential function to each participant’s data, and shaded areas indicate s.e.m. of the fit.

“9) Fig 1D hard to see difference between grey and black at first.”

Response: We now have fixed this issue in the previous Figure 1 (now **Figure 2**) and other similar figures of the revised manuscript (see [R3.9]).

“10) In Fig 6 D there are 3 columns for feature but only one is significant. If this is the informative feature, please make it clearer in the figure itself.”

Response: We thank the reviewer for pointing out this missing information. Yes, the significant column corresponds to the informative feature. We now have fixed this issue in the previous Figure 6 (now **Figure 7**) and other similar figures of the revised manuscript

(see [R3.10]).

“11) Fig 3 C. Are inhibitory neurons always mostly active really early on and then silenced or is this just in that specific trial? Also, out of curiosity, why are there no other descriptions of time courses and neural dynamics? Were they not informative?”

Response: We thank the reviewer for asking this question. Unfortunately, our selected example gave the wrong impression that the majority of inhibitory neurons were active only early on and were then silenced. In contrarily, on average, 57% of inhibitory neurons were active in each trial during the stimulus presentation period. In the revised manuscript, we now show another example to ensure that this confusion does not happen for other readers (see [R3.11]).

Regarding neural dynamic, we used representational similarity analysis to demonstrate learning-dependent changes in neural dynamics throughout the session. Nonetheless, to further elucidate the temporal dynamics of population response in our networks, we applied principal component analyses (PCA) to the response of excitatory recurrent populations of the trained RNNs because this response determines the output of the networks. More specifically, we performed three separate PCAs on the activity of excitatory recurrent populations during the simulated experiment (repeated 100 times to obtain smoother results). This includes: PCA on the response of excitatory recurrent populations to all stimuli at the beginning of each session before the network has learned about the reward environment (see Figure 10A below); PCA on the response of excitatory recurrent populations to all stimuli at the end of each session when the network has fully learned the task (see Figure 10B below); and PCA on the response of excitatory recurrent populations to all stimuli during the choice period throughout each session (see Figure 10C below). As expected, we found that the trajectory of population response projected on three principal components was not distinguishable at the beginning of the session, whereas this response diverged according to reward value as the network learned reward probabilities.

In the revised manuscript, we now have added a few sentences to discuss these results and included Figure 11 below as **Supplementary Figure 7** (see [R3.11]).

Figure 11. Dynamics of population activity in RNNs during learning task performance. (A–B) Trajectories in the activity space formed by the first three principal components of PCA performed on the response of excitatory recurrent populations at the beginning (A) and end each session (B). Diamonds mark stimulus onset, squares mark beginning of the choice period, and triangles mark the end of stimulus presentation. Different colors represent reward value (probability) assigned to each stimulus. (C) Trajectories in the activity space formed by the first three principal components of PCA performed on the response of excitatory recurrent populations during the choice period as the network learns about different stimuli. Larger markers indicate later trials within the session.

“12) Fig 4 A could use a legend in figure.”

Response: We thank the reviewer for this suggestion, which is now implemented in the previous Figure 4 (now **Figures 5**) of the revised manuscript (see [R3.12]).

“13) How would you have to change the RNN to get the feature only learning as some people (1/3) do?”

Response: We thank the reviewer for asking this question. Based on our results, we speculated that the adoption of certain strategies in our task could depend on the environments that participants experienced prior to performing our task. To test this, we considered RNNs trained in environments with very different levels of generalizability. This was possible because reward probabilities during the training step were drawn from a uniformly random distribution between 0 and 1. To that end, we defined extreme non-generalizable environments as those that fell in the 0–20% quantile of the joint distribution of features and individual features plus conjunctions generalizability indices (see Figure 12A below), and extreme generalizable environments as those that fell in the 80–100% quantile of the same joint distribution. We then examined the behavior of RNNs trained in these two types of environments.

We found that the estimated reward probabilities by RNNs trained in generalizable environments were best fit by the feature-based model (see Figure 12B below), whereas estimates by RNNs trained in non-generalizable environments were best fit by the object-based model (see Figure 12C below). These results suggest that previous exposure to environments with different levels of generalizability can result in adoption of different learning strategies as exhibited by human participants in our experiment.

We believe that investigating these mechanisms to account for the observed variability in adopted strategy is beyond the scope of this study, and thus, we did not include the above results in the revised manuscript.

Figure 12. Generalizability of the environments used during the training step influences the behavior of RNNs during the learning task. (A) The plots show the joint distribution of generalizability indices calculated for the estimated reward probabilities associated with different stimuli based on their features and combinations of individual features and conjunctions. Environments with extreme generalizability and non-generalizability used to train two different sets of RNNs are indicated with white rectangles. (B–C) The plots show the time course of explained variance (R^2) for reward probabilities estimated by RNNs using different learning models as indicated in the legend. The results for RNNs trained using extreme generalizable and non-generalizable environments are shown in B and C. Error bars represent s.e.m. The solid line is the average of exponential fits to RNNs’ data, and the shaded areas indicate s.e.m. of the fit. Different training environments result in adopting different learning strategies and different R^2 values in our task.

“14) Maybe have labels in Fig 5 (even though it wont look as nice, people might be able to follow even without remembering how the images work). Alternatively, add a legend to remind them?”

Response: We thank the reviewer for this helpful suggestion. We now have fixed this issue in the previous Figure 5 (now **Figure 6**) and other similar figures of the revised

manuscript (see [R3.13]).

“15) In the analysis of the human data, differential response for non-informative and informative features should be compared against each other if the authors want to make a comparative statement, not each against zero (this seemed to be done with feature learners but not feature and conjunction learners).”

Response: We thank the reviewer for pointing out this missing comparison, which now has been added to the revised manuscript (see [R3.14]).

“16) The authors state “In this step, only connections endowed with the plasticity mechanism were modulated after receiving reward feedback in each trial. The overall task structure used in these simulations was similar to our experimental paradigm with a simple modification where only one stimulus was shown in each trial and the network had to learn the reward probability associated with that stimulus.” This means there is no possibility for spread of effect across chosen and unchosen stimuli in the model. Was there any evidence in their participant data for such effects?”

Response: We thank the reviewer for asking this very important question. Yes, we chose this learning rule based on our experimental results. More specifically, our results of fitting participants’ choice behavior demonstrated that the mixed feature- and conjunction-based model with forgetting reward probabilities associated with the unchosen stimulus (by decaying those values toward 0.5) and the mixed feature- and conjunction-based model that does not update the unchosen options (uncoupled) both provided a better fit compared to models that allow for additional (coupled) updating of unchosen stimulus. Therefore, in our model we only consider the case for which there is no effect across chosen and unchosen stimuli.

We have now added a few sentences in the revised manuscript to clarify this point (see [R3.15]).

Reviewer #4:

“The authors studied how humans and neural networks learn values in an environment where visual features and conjunctions of features are predictive of values. They showed that both humans and a type of plastic recurrent neural networks (RNNs) showed a mixed learning strategy, in which the feature-based values are learned earlier and faster, while conjunction-based values are learned

later and slower. They showed these points with a combination of model fitting and direct data analysis. Then the authors proceeded with a detailed investigation of the RNNs. Through thorough analyses probing neural representation, connection weights, and lesioning impacts, they revealed how different neural types (plastic vs not, excitatory vs inhibitory) in the model are involved in learning the values.

This work is carefully done, with plenty of controls and complementary analyses. The paper is clearly written, and the figures are thoughtfully made. The experimental task is interesting, and the kind of recurrent neural network used is innovative. Overall, this is a very solid paper.

The main concerns I have are (1) the core finding from the human participants is, in my opinion, not so surprising, (2) the RNN part, though abundant in observations, does not provide much intuitive understanding.

I want to emphasize that these concerns do not, in any way, refute the authors' findings and observations. I'm also not suggesting major additional experiments, because I think the issue is not a lack of controls or care. The authors have done a lot of work, and in my opinion, deserve to have this paper published with few additional edits at a respectable journal."

Response: We thank the reviewer for their detailed summary and positive evaluation of our work. We hope that our new analyses, answers provided here, and corresponding changes in the revised manuscript addressed all the reviewer's remaining concerns.

Major points:

"1. Human behavior not surprising, or in other words, there was no clearly articulated alternatives to what the authors found.

The authors report that humans learn feature-based rules (red better than blue) faster than conjunction-based rules (blue triangle better than red circle). The authors emphasize that they are the first to report this phenomenon (e.g., "...our experimental results demonstrate for the first time..."). However, it's not clear what would be a plausible alternative. Could humans possibly learn conjunction-based rules before they learn feature-based rules?

This is further complicated by the fact that informative conjunction occurs much less frequently than informative features. Therefore, it's even harder to imagine than conjunctions would be learned first.

I'm willing and hoping to be convinced by the authors that this behavior is not expected by classical theories or simply our everyday intuition. But currently I don't see that from the Introduction."

Response: We thank the reviewer for pointing out the issue of not clarifying possible alternatives to what we found, making some of our experimental results look trivial. The reviewer is correct that conjunction-based learning should not be learned faster than feature-based learning because updates of feature values happen more frequently than updates of conjunction values. Nonetheless, adoption of different learning rates for feature- and conjunction-based learning can compensate for this (see below). More importantly, the most novel aspect of our behavioral results is that human participants adopted conjunction-based learning in addition to feature-based learning instead of: (1) stopping at feature-based learning; or (2) transitioning to object-based learning after the initial feature-based learning. Therefore, the observed combination of feature-based and conjunction-based learning to overcome the curse of dimensionality is not trivial at all. This is especially true because learning additional representations and using them are challenging.

To show these points more clearly, we ran additional simulations using multiple RL models to compare the accuracy of different learning strategies early in the experiment (during the first 50 trials). Specifically, we simulated RL models based on feature-based, conjunction-based, feature+conjunction based, and object-based learning with decay of reward values for the unchosen options. We found that early superiority of feature-based strategy depends on the choice of the learning and decay rates. More specifically, early in the learning, a conjunction-based model with large learning rates exhibits a smaller MSE in predicting reward probabilities than that of a feature-based learner with small learning, whereas a conjunction-based learner with small learning rates is less accurate than a feature-based learner with large learning rates (Figure 13A below). Moreover, the part of parameter space for which the feature-based learner is more accurate increases with larger decay rates (Figure 13B, C below). Similarly, an object-based learner's accuracy early in the experiment can be better than that of the best feature+conjunction (F+C₁) learner and the feature-based learner, depending on the learning and decay rates (Figure 13D-I below). Together these simulation results illustrate that dominance and thus adoption of certain learning strategies could greatly vary, making our behavioral findings non-trivial.

We have now revised the Introduction and Discussion to clarify some of these points and better explain why our behavioral findings are not trivial (see [R4.1]).

Figure 13. (A–C) Difference in the average squared error (MSE) of a feature-based learner and a conjunction-based learner in the estimation of reward probabilities during the first 50 trials of the learning task. Reinforcement learning models were simulated using the same learning rates for rewarded and unrewarded trials ($\alpha_{rew} = \alpha_{unr}$) but different values for the decay rate ($d = 0.001$ (A), $d = 0.02$ (B), $d = 0.1$ (C)) for unchosen options. The black curve indicates parameter values for which the difference is equal to 0 corresponding to similar precision of feature-based and conjunction-based learners. (D–F) Same as (A–C) but comparing object-based and F+C₁ learners. (G–I) Same as (A–C) but comparing object-based and feature-based learners.

“2. RNN results lack a summarizing intuition.

The RNN itself is interestingly complex in its architecture (two cell types, optional input plasticity, optional recurrent plasticity), and the authors conducted a very thorough investigation of it.

However, at the end of it, we are left with lots of observations of the RNN, but little intuition that

may transfer to other settings.

For example, the authors observed that feature-encoding input units connect more strongly to inhibitory neurons that have plastic input weights. Why? As far as I can tell, little intuition is provided, and little investigation is oriented at understanding the difference between cell types, rather than reporting it. Some sections describing the RNN results (parts of p12-p19) almost read like auto generated text reporting all combinations of analyses (e.g. X analysis on Y cell report Z value).

I think this issue is demonstrated in the authors' Abstract where they say the "learning strategy relies on...distinct contributions of inhibitory and excitatory neurons". That sounds rather vague."

Response: We thank the reviewer for raising this issue. The reviewer is correct that we did not provide clear intuition for some of the simulation results and especially the opposite roles of inhibitory and excitatory neurons in the adoption of different learning strategies. Indeed, determining why and how the opponency between representations of object and feature values is materialized in the excitatory and inhibitory populations, respectively, was one of the main goals of the different simulations and analyses we performed.

Because of the observed transition between different types of learning strategies, we expected that representations of feature and object values should compete. On the one hand, the ability of excitatory recurrent populations to influence the output population could explain the larger contribution of excitatory populations to the object-based strategy. More specifically, object values can be estimated directly by a single connection from sensory populations to recurrent populations because they do not require integration of information across features and/or conjunctions as is the case for feature-based and conjunction-based strategies. As a result, object values could directly drive excitatory populations which in turn drive the output of the network.

However, another important factor in driving this competition or opponency is the interaction between excitatory and inhibitory neurons. This interaction allows for value-based modulation of the excitatory populations by the inhibitory populations whose value representations are more sensitive to different learning strategies (e.g., compare **Figure 6G** and **Figure 6C**). Specifically, our analysis of naïve input weights suggests that feature-based strategy is mainly encoded in the input connections from feature-encoding sensory populations to inhibitory populations. Additionally, because of recurrent connections, inhibitory populations disinhibit excitatory populations according to a feature-based strategy, and because our learning rules depend on pre- and post-synaptic activity, learning in input connections from sensory populations to recurrent populations becomes

dominated by the feature-based and not object-based strategy. Therefore, because of this value-dependent disinhibition, representations of feature values are reinforced (while suppressing object-based values) in excitatory populations, allowing for the intermediate strategies to emerge.

Consistently, we found that the rate of value-dependent changes in the input connections from object-encoding sensory populations to recurrent connections is not significant in the intact network (**Figure 7D**). However, once the connections from inhibitory population are lesioned, feature-based disinhibition is removed and the object-based strategy dominates (**Figure 8**). Therefore, the results of lesioning connections from inhibitory to excitatory populations support the idea that interactions between recurrent populations are needed to suppress object-based strategy and for the emergence of mixed learning strategy.

In summary, our multiple analyses show that differential connections of feature-encoding and object-encoding sensory neurons to excitatory and inhibitory populations results in an opponency between representations of feature and object values. This opponency by excitatory and inhibitory neurons allows for value-based modulation of the excitatory neurons through the inhibitory neurons and enables the adoption of intermediate strategies.

To provide more intuition, we now have revised the discussion of the RNNs' results, included some of the points above to the Discussion, and clarified the distinct contribution of excitatory and inhibitory neurons in the Abstract (see [R4.2]).

"3. Related to the second point, the architecture comparison is not given enough attention. Toward the end of the Results section, the authors discussed what happens with alternative structures. I find this section with lots of promises, because it may help us understand why the plastic RNN is necessary and why the human findings are surprising. In particular, I'm surprised to know that some architectures didn't learn like humans do, which can be an important reason to think that the human findings are not trivial (essentially countering my first major concern). But being a short section at the very end, it feels like an afterthought and a missed opportunity. If I assumed the authors had infinite time and patience, I would have suggested that they restructure the paper, and put more emphasis on this last section. Perhaps bring it up, and show it before the main RNN results. But I know that in reality, we all have many things to do, and it's not worth restructuring a perfectly fine paper just to satisfy one reviewer's preference. So I'm not suggesting the authors do anything about this. I'm only pointing this issue out in the hope that the authors may find it helpful for their future work."

Response: We thank the reviewer for their suggestion on emphasizing on the results of alternative models and also their understanding of the time it takes to restructure the manuscript. We think that our changes in response to their previous comments (R4.1 and R4.2) should have made our experimental results look less trivial and also provided more clear intuition for simulation results with intact network. Nonetheless, we also have revised parts of the Introduction and Discussion to better emphasize behavior of RNNs with alternative architectures that behave very differently from our human participants (see [R4.3]).

“Minor points

4. In the Abstract, it’s not always clear what findings are based on RNNs and what are from humans.”

Response: This has been clarified in the revised manuscript (see [R4.4]).

“5. Fig. 1: I would appreciate if there’s a schematic for the task. Why not move Fig. S1 A, B to Fig. 1?”

Response: We thank the reviewer for their helpful suggestion on including the task description in the main text. In the revised manuscript, we have moved previous Supplementary Figure 1 to the main text as **Figure 1** (see [R4.5]).

“6. Fig. 2 and page 8, the effect for informative conjunction seems relatively weak ($P=0.029$). But I suppose it’s not clear if the authors can do anything about it.”

Response: Analyzing estimates reported by human participants, we demonstrated that they learn the informative conjunction at a slower rate compared to the informative feature (see Figure 14A below). Consequently, this predicts that differential response for the informative conjunction will be small in magnitude, explaining the weakness of this effect (see Figure 14B below). Nonetheless, to investigate whether the strength of the effect depends on the time point in the experiment, we also analyzed data from the second half of each session (trials [216–432]) where conjunction-based learning was more pronounced. Limiting our differential response calculation to those trials, we found a stronger effect (two-sided sign-rank test; $P = 0.005$, $d = 0.41$, $N = 41$) (see Figure 14C below). Nonetheless,

considering that such analysis would discard half of the participants' choice sequences, we kept the calculations of differential response as is, using all choice trials.

Figure 14. (A) Time course of adopted learning strategies measured by fitting participants' estimates of reward probabilities. Plotted is the weight of the informative feature and informative conjunction in the F+C1 model. Error bars indicate s.e.m. The solid line is the average of fitted exponential function to each participant's data, and shaded areas indicate s.e.m. of the fit. (B–C) Plot shows the histogram of differential response calculated using all choice sequences (B), and choice sequences in the second half of the session (C) for the informative conjunction in participants whose choice behavior was best fit by the F+C1 model. The dashed lines show the median values across participants, and an asterisk indicates the median is significantly different from 0 using two-sided sign-rank test with $P < 0.05$.

"7. The reason for using RNN is not well articulated. Usually, people use RNNs to have temporal dynamics, but Fig. 3c shows essentially no dynamics?"

Response: We thank the reviewer for pointing out missing analyses of RNNs' dynamics. Although single trial temporal dynamics seem simple in our task, the networks still needed to demonstrate long-term complex dynamics when learning about different stimuli in the task. Our RSA analysis is a great demonstration of how learning in our task results in across-trial dynamics in RNNs that are dependent on the reward structure in the task.

To further confirm this, we applied principal component analyses (PCA) to the response of excitatory recurrent populations of the trained RNNs because this response determines the output of the networks. More specifically, we performed three separate PCAs on the activity of excitatory recurrent populations during the simulated experiment (repeated 100 times to obtain smoother results). This includes: PCA on the response of excitatory recurrent populations to all stimuli at the beginning of each session before the

network has learned about the reward environment (see Figure 15A below); PCA on the response of excitatory recurrent populations to all stimuli at the end of each session when the network has fully learned the task (see Figure 15B below); and PCA on the response of excitatory recurrent populations to all stimuli during the choice period throughout each session (see Figure 15C below). As expected, we found that the trajectory of population response projected on three principal components was not distinguishable at the beginning of the session, whereas this response diverged according to reward value as the network learned reward probabilities.

In the revised manuscript, we now have added a few sentences to discuss these results and included Figure 15 below as **Supplementary Figure 7**. Additionally, we have clarified the reason for using RNNs (see [R4.6]).

Figure 15. Dynamics of population activity in RNNs during learning task performance. (A–B) Trajectories in the activity space formed by the first three principal components of PCA performed on the response of excitatory recurrent populations at the beginning (A) and end of each session (B). Diamonds mark stimulus onset, squares mark beginning of the choice period, and triangles mark the end of stimulus presentation. Different colors represent reward value (probability) assigned to each stimulus. (C) Trajectories in the activity space formed by the first three principal components of PCA performed on the response of excitatory recurrent populations during the choice period as the network learns about different stimuli. Larger markers indicate later trials within the session.

“8. Fig. 7. Perhaps write something like “Lesioned networks” in the figure (not just the caption)?”

Response: Thanks for this useful suggestion which now has been incorporated in **Figure 8** and similar figures of the revised manuscript (see [R4.7]).

REVIEWER COMMENTS

Reviewer #1 (Remarks to the Author):

I thank the authors for their responses to my comments. I am satisfied by their responses. My only comment is, I would suggest adding more information about how RNN was simulated [regarding points 2,3] and how the final choice layer was trained.

Reviewer #2 (Remarks to the Author):

The authors have addressed my comments. I have no further concerns.

Reviewer #3 (Remarks to the Author):

The authors addressed all my comments.

Reviewer #4 (Remarks to the Author):

I'd like to thank the authors for carefully responding to my suggestions. The authors added several new analyses and made lots of changes to the manuscript. Overall, I'm happy with the changes.

I think my major concern #1 was addressed. My major concern #2 was alleviated.

Minor:

I would generate Figure 13 with a "divergent" colormap, such as BwR in matplotlib.

I appreciate the clever numbering of responses (e.g., [R4.5]). That's smart!

Response to the final reviewers' comments, and summary of changes made in response to the comments of the reviewers.

Title: Computational mechanisms of distributed value representations and mixed learning strategies

Authors: Farashahi and Soltani

We are greatly thankful to all the reviewers and the editor for their careful reading of our revised manuscript. Below, please see how we have addressed their final concerns.

Reviewer #1

I thank the authors for their responses to my comments. I am satisfied by their responses. My only comment is, I would suggest adding more information about how RNN was simulated [regarding points 2,3] and how the final choice layer was trained.

Response: We have addressed these issues in the final revision of our manuscript as follows:

(page 32) "Specifically, without retraining the networks, we added a decision layer after the output layer (using a logistic function) to generate binary choice between pairs of options and fit simulated choice data (similar to our participants) to show our results and main conclusion do not depend on the choice mechanism."

Reviewer #4

I'd like to thank the authors for carefully responding to my suggestions. The authors added several new analyses and made lots of changes to the manuscript. Overall, I'm happy with the changes.

I think my major concern #1 was addressed. My major concern #2 was alleviated.

Minor:

I would generate Figure 13 with a "divergent" colormap, such as BwR in matplotlib.

I appreciate the clever numbering of responses (e.g., [R4.5]). That's smart!

Response: We have now revised Figure 13 (see below) in the rebuttal to follow this suggestion and included it as a Supplementary Figure 3 and discussed its results in the revised manuscript.

Supplementary Fig. 3. Comparison of estimation error between different models. (a–c) Plots show the difference in the average squared error (MSE) of a feature-based learner and a conjunction-based learner in the estimation of reward probabilities during the first 50 trials of the learning task. Reinforcement learning models based on feature-based, conjunction-based, mixed feature- and conjunction-based, and object-based learning were simulated using the same learning rates for rewarded and unrewarded trials ($\alpha_{\text{rew}} = \alpha_{\text{unr}}$) but different values for the decay rate ($d = 0.001$ (a), $d = 0.02$ (b), $d = 0.1$ (c)) for unchosen options. The black curve indicates parameter values for which the difference is equal to 0 corresponding to similar precision of feature-based and conjunction-based learners. (d–f) Same as (a–c) but comparing object-based and F+C₁ learners. (g–i) Same as (a–c) but comparing object-based and feature-based learners.